# ARROW: Accelerator for Time Series Causal Discovery with Time Weaving

Yuanyuan Yao [1]   Yuan Dong [1]   Lu Chen[✉ 1]   Kun Kuang [1]   Ziquan Fang [1]
Cheng Long [2]   Yunjun Gao [1]   Tianyi Li [3]

## Abstract

Current causal discovery methods for time series data can effectively address a variety of scenarios; however, they remain constrained by inefficiencies. The significant inefficiencies arise primarily from the high computational costs associated with binning, the uncertainty in selecting appropriate time lags, and the extensive sets of candidate variables. To achieve both high efficiency and effectiveness of causal discovery, we introduce an accelerator termed ARROW. It incorporates an innovative concept termed "Time Weaving" that efficiently encodes time series data to well capture the dynamic trends, thereby mitigating computational complexity. We also propose a novel time lag discovery strategy utilizing XOR operations, which derives a theorem to obtain the optimal time lag and significantly enhances the efficiency using XOR operations. To optimize the search space for causal relationships, we design an efficient pruning strategy that intelligently identifies the most relevant candidate variables, enhancing the efficiency and accuracy of causal discovery. We applied ARROW to four different types of time series causal discovery algorithms and evaluated it on 25 synthetic and real-world datasets. The results demonstrate that, compared to the original algorithms, ARROW achieves up to 153x speedup while achieving higher accuracy in most cases.

## 1. Introduction

Learning the causal relationships on multivariate series holds significant practical value in the field of time series mining (Assaad et al., 2022; Zhou & Chen, 2022;

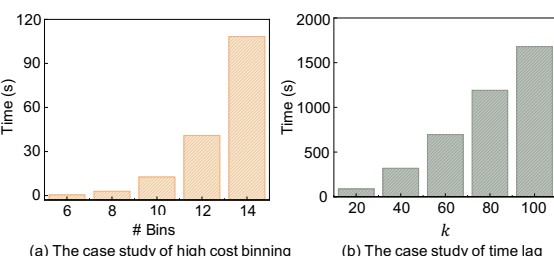

*Figure 1.* Case Study

Yao et al., 2024c). Causal discovery not only deepens our understanding of dynamic relationships among multiple variables (Zhao et al., 2023) but also provides scientific evidence to solve practical problems (Castro et al., 2023; Martínez-Sánchez et al., 2024; Huang et al., 2019). For example, in Alzheimer's disease research, exploring causal interactions between brain regions can help reveal pathological mechanisms and optimize treatment strategies (Liu et al., 2024). Furthermore, the study of causal relationships can assist downstream tasks (e.g., forecasting, anomaly detection, etc.) (Qiu et al., 2024; 2025; Yi et al., 2024; Xia et al., 2025; Yao et al., 2023; 2024b). By discovering causal relationships within multivariate time series, deeper feature information can be provided, considerably enhancing the performance and accuracy of downstream tasks. This causal-driven analysis method is becoming one of the core tools for solving complex time series problems. Recent research in the field of time series causal discovery has effectively revealed causal relationships under different conditions. However, through case studies conducted on various methods and datasets, we found that significant bottlenecks still exist in terms of efficiency due to the following limitations:

- **High Computational Costs of Binning.** Currently, most causal discovery methods (Runge, 2018; Runge et al., 2019; Martínez-Sánchez et al., 2024) rely on statistical techniques such as conditional independence testing and mutual information, often using discretization (i.e., binning) of continuous numerical data for analysis. A major issue with this approach is that the computational complexity of independence testing increases significantly as the binning granularity becomes finer, as shown in Figure 1(a), especially when dealing with large-scale time series data. Additionally, the above binning-based methods struggle to capture dynamic causal structures.

[1]The College of Computer Science, Zhejiang University, Hangzhou 310027, China [2]College of Computing and Data Science, Nanyang Technological University, Singapore [3]The Department of Computer Science, Aalborg University, Denmark. Correspondence to: Lu Chen <luchen@zju.edu.cn>.

- **Uncertainty in Appropriate Time Lag Selection.** The selection of time lag has a significant impact on the results of most causal discovery methods (Hyvärinen et al., 2008). However, current methods (Martínez-Sánchez et al., 2024; Gerhardus & Runge, 2020) usually rely on brute-force search, where all possible time lag values are traversed to construct the best summary graph. This approach has clear drawbacks: first, the Brute-Force search results in an unacceptable computational time when dealing with large-scale time series data as shown in Figure 1(b); second, it lacks a theoretical mechanism to determine the optimal time lag to generate the best summary graph, thus making the results less convincing and reliable. For example, when applying the *PCMCI* algorithm to a dataset with a time lag of 6, identifying the correct time lag can achieve an accuracy of up to 0.99. However, an incorrect time lag 5 may result in an accuracy drop of 49%.

- **Extensive Sets of Candidate Variables.** Current causal discovery algorithms (Martínez-Sánchez et al., 2024; Tank et al., 2021) typically utilize different combinations of all variables to form candidate sets. As the number of variables increases, the size of the candidate parent node set grows exponentially. For instance, given $d$ variables, $2^d$ tests are required; if $T$ time lag needs to be determined, the number of tests increases to $2^{dT}$. The dimensionality explosion of the candidate set can compromise the accuracy of causal discovery by obscuring true causal relationships (Runge et al., 2019).

**Our approach:** To address these limitations, we propose a time series causal discovery accelerator called ARROW. This accelerator can seamlessly integrate with most time series causal discovery algorithms, significantly improving efficiency without compromising the performance of the original algorithm. The key idea is to capture the contextual trends of a value in a binary tuple. Based on the binary representation, we can efficiently and effectively extract time lags and uncover causal relationships.

We first introduce time weaving, an innovative method for encoding individual time points. This method divides the data into the smallest units with contextual information by considering the trend changes between three consecutive points in the time series; each unit represents the trend between these three points. Here, we use 1 bit to represent the trend between two consecutive points, where 1 indicates an upward trend and 0 represents a downward trend. Based on the time-weaving encoding, we analyze the patterns of trend changes between variables using XOR operations and derive a theorem to determine the optimal time lag (Yao et al., 2024a). Specifically, when determining time lag, we first rank the potential causal relationships between variables, and then select a representative subset based on the ranking results. Finally, we propose an efficient pruning strategy. This strategy optimizes the causal relationship search by

intelligently selecting the most promising candidate set, rather than considering all variables in the candidate parent node set. By focusing on the variables that contribute most to causal discovery, we can effectively reduce the size of the search space, improving the efficiency and accuracy of causal relationship discovery. We applied ARROW to four different types of causal discovery algorithms. Experimental results on 25 synthetic and real-world datasets demonstrate that ARROW significantly enhances efficiency while maintaining the original performance of the algorithms.

## 2. Related work

**Time series causal discovery algorithms.** Mainstream time series causal discovery methods include constraint-based (Runge et al., 2019; Runge, 2020; Entner & Hoyer, 2010; Gerhardus & Runge, 2020), score-based (De Campos & Ji, 2011; Gao et al., 2022; Pena et al., 2005; Hyvärinen et al., 2010), granger-based (Tank et al., 2021; Xu et al., 2019; Marcinkevičs & Vogt, 2021; Wang et al., 2018; Xu et al., 2016), and information-theoretic (Martínez-Sánchez et al., 2024; Chaves et al., 2015) approaches. Constraint-based methods constrain the causal structure using conditional independence tests, relying on statistical and conditional independence to infer causal relationships by removing non-directly related edges and applying directional rules (such as d-separation (Liu et al., 2024)). Score-based methods quantify model fit and complexity by defining scoring functions (Neath & Cavanaugh, 2012; Burnham & Anderson, 2004) and use search strategies (such as greedy search) to find the optimal directed acyclic graph (DAG) in the model space. In contrast, Granger-based methods utilize the lag effect in time series data, determining if one variable significantly explains the changes in another variable through linear regression or forecasting performance, thus inferring causal relationships. Information-theoretic methods infer causal relationships by calculating the information gain or mutual information between variables, offering advantages in handling nonlinear relationships, multivariate dependencies, and robustness to noise. While these four types of methods can effectively address causal discovery under different conditions, they remain constrained by inefficiencies. We aim to propose a general acceleration solution that can address the current efficiency issues in causal discovery.

**Acceleration strategy.** Current acceleration methods are mainly based on hardware acceleration, including CPU-based (Le et al., 2016; 2018; Madsen et al., 2015; Schmidt et al., 2019), GPU-based (Zarebavani et al., 2019; Hagedorn & Huegle, 2021; Schmidt et al., 2018; Hu et al., 2021), and FPGA-based (Guo & Luk, 2022; Guo et al., 2023b;a) approaches. For instance, ParallelPC (Le et al., 2016; 2018) is the first method to implement parallel causal discovery, where the core idea is to execute conditional independence

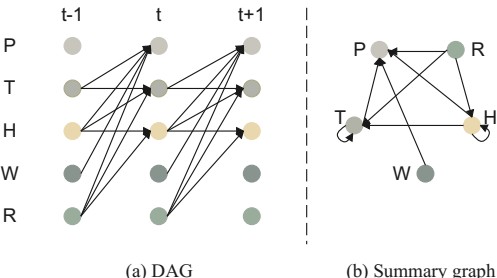

(a) DAG          (b) Summary graph

*Figure 2.* An example of DAG and Summary graph

(CI) tests in parallel. cuPC (Zarebavani et al., 2019) is the first method to accelerate causal discovery using GPUs, and it employs a lossless transformation of the PC-stable algorithm with two optimizations: first, if a CI test between a pair of variables returns true, cuPC immediately cancels all remaining tests; second, CI tests with the same conditioning set are executed within the same local thread, reducing thread management costs. gpuPC (Hagedorn & Huegle, 2021) is a GPU-based tool designed for discrete data, and its CI tests require auxiliary data structures (e.g., contingency tables), which consume substantial GPU memory. CSF (Guo & Luk, 2022) is the first FPGA-based acceleration method for constraint-based causal discovery, effectively utilizing FPGA on-chip storage and parallel processing capabilities while reducing communication costs. The CSF implementation achieves higher speed than cPU-based and GPU-based approaches. These hardware acceleration methods primarily address efficiency issues in CI (Runge, 2018; Spirtes et al., 2001) independence testing, but they are not universally applicable, especially for score-based and Granger-based methods. In contrast, the acceleration method we propose is based on data-level optimization, making it more general and applicable to all causal discovery methods. It can also be combined with hardware acceleration methods to further enhance efficiency.

## 3. Preliminaries and Background

A Structural Causal Model (SCM) is a graphical representation that illustrates causal relationships, capturing how interventions on one or more variables affect the values of other variables within the data generation process (Pearl, 2009; Gong et al., 2023). Formally, an SCM is represented as a 4-tuple $(V, U, F, P(U))$, where $V$ and $U$ denote the sets of endogenous and exogenous variables, respectively, $P(U)$ represents the distribution of exogenous variables, and $F$ is the set of mapping functions. Specifically, each endogenous variable is formally defined by the model $x_i := f_i(\mathbf{PA}(x_i)), i = \{1, \ldots, |V|\}$, where $|V|$ denotes the number of endogenous variables, while $\mathbf{PA}(x_i)$ represents the set of parents of $x_i$ and $\mathbf{PA}(x_i) \subseteq U \cup V$. The function $f_i \in F$ denotes a function between its structural parents $\mathbf{PA}(x_i)$.

For each SCM, we can construct a directed acyclic causal graph (DAG) $G$ by associating a vertex with each $x_i$ and directing edges from each parent variable in $\mathbf{PA}(x_i)$ (the causes) to the corresponding child $x_i$ (the effect).

In the causal graph of SCM, each variable is conditionally independent of its non-effects, given its direct causes (Pearl, 2009). In other words, a variable is independent of variables that do not directly influence it, once its parents (i.e., direct causes) are known. This principle is crucial for causal discovery as it allows the identification of causal effects from observational data. Formally, the causal Markov condition asserts that the joint distribution can be factorized as:

$$P(\mathbf{x}) = \prod_i^{|V|} P(x_i \mid \mathbf{PA}(x_i))$$

We study the problem of causal discovery on time series. Thus, we first define the multivariate time series below.

**Definition 3.1 (Multivariate Time Series).** A multivariate time series (MTS) $\mathbf{X}$ with $d$ variables is defined as: $\mathbf{X} = \{\mathbf{x}^t\}_{t=0}^{T-1} = \{(x_1^t, x_2^t, \ldots, x_d^t)^\top\}_{t=0}^{T-1}$, where $t$ represents a discrete time point, $T$ is the total length of $\mathbf{X}$ in the time dimension, and each $\mathbf{x}^t$ is a $d$-dimensional vector $(x_1^t, x_2^t, \ldots, x_d^t)$ at time point $t$.

**Assumption 3.2 (Markov Property).** The Markov property of time series assumes that the future slice $\mathbf{x}^{t+1}$ depends on the current state $\mathbf{x}^t$ but does not depend on the history $(\mathbf{x}^1, \ldots, \mathbf{x}^{t-1})$, i.e, $P(\mathbf{x}^{t+1} \mid \mathbf{x}^t, \mathbf{x}^{t-1}, \ldots, \mathbf{x}^1) = P(\mathbf{x}^{t+1} \mid \mathbf{x}^t)$ (Figueiredo et al., 2018).

**Definition 3.3 (Time Lag).** Causal effects do not occur instantaneously, but rather unfold gradually over time for MTS (Peters et al., 2013). Given $\mathbf{x}_i^t = f_i(\mathbf{x}_j^{t-k}, u_i)$, where $k$ represents the time lag, indicating that the impact of $\mathbf{x}_i$ on $\mathbf{x}_j$ is delayed by $k$ time points.

Combined with the Assumption 3.2, we focus on exploring causality in MTS with a time lag $k$, where the current point $\mathbf{x}^t$ is influenced by the value $\mathbf{x}^{t-k}$. Notably, different pairs of variables may exhibit distinct time lags in some cases.

**Definition 3.4 (Summary Graph).** Let $G := (V, E)$ be the associated summary graph of an SCM model (Peters et al., 2013). The vertex set $\mathbf{V} := \mathbf{X}$, the edge set $\mathbf{E}$ contains $\mathbf{x}_i \to \mathbf{x}_j (i \neq j)$ iff $\mathbf{x}_i \in \mathbf{PA}(\mathbf{x}_j)$.

The primary goal of time series causal discovery is to construct a summary graph that intuitively represents the causal relationships between different variables.

**Definition 3.5 (Causal Discovery from MTS).** Given a MTS data $\mathbf{X}$, assume that causal relationships between variables are given by the following structural equation model: $x_i(t) := f_i(\mathbf{PA}(x_i^{t-k}), N_i), i = \{1, \ldots, d\}, 0 \leq k < T$, $\mathbf{PA}(x_i^{t-k})$ is the set of direct parents of $x_i^{t-k}$, $N_i$ denotes the independent noise and can represent either measurement

noise or driving noise without losing generality. Causal discovery from MTS aims to find the Summary Graph.

Note that, the summary graph can be denoted as an adjacency matrix $A \in \mathbb{Z}^{d \times d}$. The $(i, j)$-th entry of the matrix $A$ is 1 if past observations of $x_i$ affect $x_j^t$, and 0 otherwise. We say that '$x_i$ causes $x_j$' if $A_{ij} = 1$.

**Example 1.** We propose an SCM for air quality causal discovery. The model includes three endogenous variables (i.e., air pollution concentration $P$, temperature $T$, and humidity $H$), and two exogenous variables (wind speed $W$ and precipitation $R$). The model can be represented as a 4-tuple $SCM = \langle V, U, F, P(U) \rangle$, where $V = \{P, T, H\}$ are the endogenous variables, $U = \{W, R\}$ are the exogenous variables, $F$ represents the mapping functions, and $P(U)$ is the joint distribution of exogenous variables. The air pollution concentration $P$ is influenced by temperature, humidity, wind speed, and precipitation, with the functional relationship given by $P_t := f_P(T_{t-1}, H_{t-1}, W_{t-1}, R_{t-1})$; The temperature $T$ is influenced by its own previous value and humidity, expressed as $T_t := f_T(T_{t-1}, H_{t-1})$; The humidity $H$ is affected by temperature, its own previous value, and precipitation, represented as $H_t := f_H(H_{t-1}, R_{t-1})$. The causal relationships based on SCM are illustrated in the following directed acyclic graph (DAG) as shown in Figure 2(a), where each directed edge indicates the causal dependency between variables at different time points.

The summary graph depicted in Figure 2(b) represents a condensed version of the directed acyclic graph (DAG) (Vowels et al., 2022) shown in Figure 2(a). In this summary graph, the vertices correspond to the variables, while the edges capture the direct causal dependencies between them.

**Our problem.** The main focus of our paper is not on the causal discovery algorithms themselves but on addressing the efficiency challenges in MTS causal discovery, caused by the high computational cost of binning, the uncertainty in time lag selection, and the extensive sets of candidate variables. Different from the hardware-level acceleration, we aim to develop **a data-level acceleration framework** that enhances the efficiency of any causal discovery algorithm while preserving its accuracy.

## 4. ARROW Method

We proceed to detail our adaptive accelerator ARROW, including three components (i.e., time weaving, time lag discovery, and candidates pruning) as shown in Figure 3.

### 4.1. Time Weaving

For two variables with a causal relationship, a change in the value of one variable will directly or indirectly cause a change in the value of the other variable after a certain

time delay. This change in value effectively reflects the variable's trend. Based on this observation, we propose a new concept termed Time Weaving to capture the local dynamic characteristics between time points. It encodes a specific time point via the trend changes of its preceding and succeeding points. This approach preserves the core characteristics of individual time points while integrating the local contextual relationships within the time series. We formally define Time Weaving below.

**Definition 4.1 (Time Weaving).** Given a MTS data with $d$ variables, denoted as $\mathbf{X} = \{\mathbf{x}^t\}_{t=0}^{T-1} = \{(x_1^t, x_2^t, \ldots, x_d^t)^\top\}_{t=0}^{T-1}$. We introduce a weaving window $w$ and represent $x_i^t$ ($1 \le i \le d$ and $w \le t \le T - w$) using its trend $tr$ relative to the values $x_i^{t-w}$ and $x_i^{t+w}$ at the $(t-w)$-th and $(t+w)$-th timestamps, i.e., $x_i^t$ can be encoded as $(tr_i\langle t-w, t\rangle, tr_i\langle t, t+w\rangle)$. Specifically, the trend $tr_i\langle t, t'\rangle$ equals to 1 if $x_i^t > x_i^{t'}$, otherwise $tr_i\langle t, t'\rangle = 0$, where $t$ and $t'$ are arbitrary time points. Therefore, the time weaving value $\mathbf{X}_{tw}$ of $\mathbf{X}$ is defined as a list of tuples:

$$\mathbf{X}_{tw} = \{(\mathbf{tr}\langle t-w, t\rangle, \mathbf{tr}\langle t, t+w\rangle)\}_{t=w}^{T-w}, \quad (1)$$

where $\mathbf{tr} = (tr_1, \ldots, tr_d)$.

**Example 2.** Given a MTS data $\mathbf{X}$ with $d = 3$ variables and a sequence length of $T = 5$ as shown in the Figure 3(a):

$$\mathbf{X} = \begin{bmatrix} 0.23 & 2.89 & 2.01 & 2.31 & 1.89 \\ 1.01 & 1.78 & 1.21 & 0.35 & 1.31 \\ -0.12 & -0.88 & 0.13 & 1.37 & 0.03 \end{bmatrix},$$

Assume that the weaving window $w = 1$. For the value $x_1^1 = 2.89$ at $t = 1$ in the first variable, its trend encoding relative to $t - w = 0$ and $t + w = 2$, and thus, it can be encoded as $(1, 0)$, as $tr_1\langle 0, 1\rangle = 1$ and $tr_1\langle 1, 2\rangle = 0$. Similarly, for the value $x_2^1 = 1.78$, its trend weaving value is $(1, 0)$, as $tr_2\langle 1, 2\rangle = 1$ and $tr_2\langle 2, 3\rangle = 0$. Thus, the time weaving representation of $\mathbf{X}$ at $t = 1$: $\mathbf{X}_{tw}^0 = ((1, 0), (1, 0), (0, 1))^\top$. The overall $\mathbf{X}_{tw}$ of $\mathbf{X}$ is:

$$\mathbf{X_{tw}} = \begin{bmatrix} (1, 0) & (0, 1) & (1, 0) \\ (1, 0) & (0, 0) & (1, 0) \\ (0, 1) & (1, 1) & (0, 1) \end{bmatrix}.$$

Time weaving is the key component of the causal discovery of the time series accelerator termed ARROW. In this way, we transform the values of the time series into tuples composed of binary 0 and 1. This transformation not only discretizes continuous time series values into 0 and 1, eliminating uncertainties in bucketization but also enhances the consistency and efficiency of data processing. Furthermore, after the transformation of the time-weaving, the data incorporates contextual information, no longer representing a single value, which improves the effectiveness of causal discovery. The time weaving format can be directly used for downstream causal discovery algorithms.

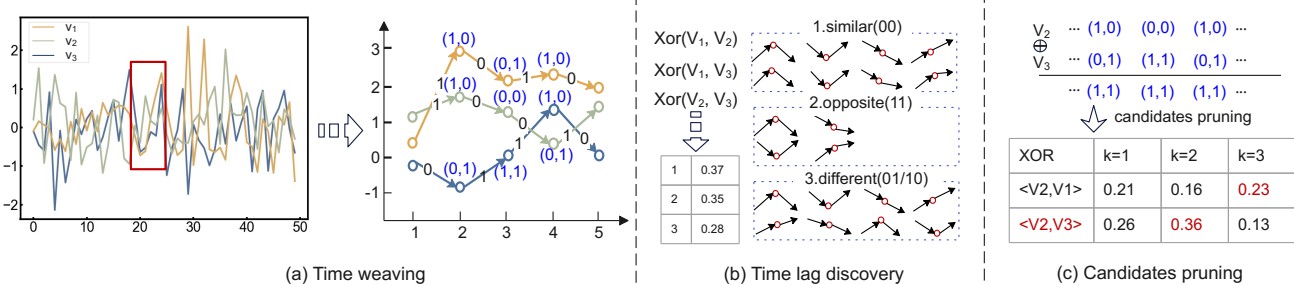

*Figure 3.* The overview of ARROW

## 4.2. Time Lag Discovery

We observe that if two variables exhibit a significant causal relationship at a time delay of $k$, their individual trends at specific time points must align or oppose each other. Based on this observation, we need to address two key questions.

- Q1: How to define the relationship between their trends?
- Q2: How to use the trends to identify the time lag?

To address Q1, we first define the trend of a point $v^t$ using the concept of time-weaving, categorizing it into three distinct states: $(0,0)$, $(1,1)$, and $(1,0)/(0,1)$. As illustrated in Figure 3(b), the relationship between two points can be described by one of ten possible pairwise combinations: $(0,0)$-$(0,0)$, $(0,0)$-$(0,1)$, $(0,0)$-$(1,0)$, $(0,0)$-$(1,1)$, $(0,1)$-$(0,1)$, $(0,1)$-$(1,0)$, $(0,1)$-$(1,1)$, $(1,0)$-$(1,0)$, $(1,0)$-$(1,1)$, and $(1,1)$-$(1,1)$. Here, the pairwise combination is symmetric, i.e., $(0,0)$-$(1,0)$ equals to $(1,0)$-$(0,0)$. When the trends of the two points are either fully aligned or completely opposite—such as $(0,0)$-$(0,0)$, $(0,0)$-$(1,1)$, $(0,1)$-$(0,1)$, $(0,1)$-$(1,0)$, or $(1,0)$-$(1,0)$—their XOR operation will yield results like '00 ∧ 00 = 00' or '10 ∧ 01 = 11', which are always either '00' (indicating similarity) or '11' (indicating opposition). On the other hand, if the trends are only partially aligned—for example, $(0,0)$-$(0,1)$, $(0,0)$-$(1,0)$, or $(1,1)$-$(1,0)$—their XOR operation produces results like '01 ∧ 00 = 01' or '00 ∧ 10 = 10', which are either '01' or '10' (indicating a difference). Therefore, the relationship between two points can be classified into three distinct categories based on their XOR results:

- *Similar* $(0,0)$ : The trends are fully aligned.
- *Opposite* $(1,1)$: The trends are completely reversed.
- *Different* $((0,1)$ or $(1,0))$: The trends show partial alignment or deviation.

It solves Q1, on how to define the trend relationship $\mathbf{Xor}(v, v', k)$ between two variables $v$ and $v'$ in a time lag $k$, where $k$ indicates the impact of $v$ on $v'$ is delayed by $k$ timestamps (cf. Def. 3.3). Here, $\mathbf{Xor}(v, v', k)$ at time $t$ is computed via the XOR operation on time-weaving representations of $v^{t-k}$ and $v'^{t}$.

To further address Q2, we propose a novel time lag dis-

covery strategy aimed at quickly and accurately identifying the specific range of time lags. This strategy effectively detects time delays in causal relationships, helping to reveal complex dynamic interactions.

**Theorem 4.2.** *Given two variables $v$ and $v'$ in a MTS data* $\mathbf{X}$, *if $v$ and $v'$ have a causal relationship in a time lag $k$, then either* $P(\mathbf{Xor}(v, v', k) = (0,0))$ *or* $P(\mathbf{Xor}(v, v', k) = (1,1))$ *will exceed* $\frac{1}{3} - \epsilon$, *where $\epsilon$ is a value approaching 0.*

The proof is in Appendix B. This addresses Q2, on how to use the trends to identify the optimal time lag. In other words, the optimal $k$ value makes either $P(\mathbf{Xor}(v, v', k) = (0,0))$ or $P(\mathbf{Xor}(v, v', k) = (1,1))$ larger than $\frac{1}{3} - \epsilon$ for any variable pairs with the causal relationship (we set 0.33 in our method). Leveraging the XOR results to determine causal relationships between variables is highly efficient due to the use of bitwise operations. Additionally, by integrating the joint distribution with the XOR outcome, this approach accurately identifies optimal time lags, offering an effective and efficient method for uncovering causal relationships.

## 4.3. Candidates Pruning

Although the most likely time lag between each pair of variables can be inferred, if there is no causal relationship between $v$ and $v'$, the inferred time lag is likely to be incorrect. Therefore, to minimize such errors, candidates pruning of the variables is necessary. Algorithm 1 is designed to filter significant variable pairs based on their time-lagged relationships in a MTS data. It takes two inputs: $\mathbf{X}_{tw}$, the time weaving representations of $\mathbf{X}$, and *Thr*, a pruning threshold for the percentage of valid variable pairs. The output is a set of filtered variable pairs, denoted as $C$.

**Step 1: Time Lag Discovery.** We first iterate over all possible variable pairs $(v, v')$ from the $d$ variables. For each pair, it evaluates all potential time lags $k$ ranging from 1 to $T - \omega + 1$, where $T$ is the time series length and $\omega$ is the window size. For each possible lag value $k$, it computes the XOR operation between two variables to identify patterns in their binary representation. It then calculates the percentages of specific binary tuple occurrences—$(0,0)$, $(1,1)$, and $(0,1)/(1,0)$—and stores them as $R0[v, v', k]$, $R2[v, v', k]$,

---

**Algorithm 1** Candidates Pruning

---

**Input:** Time-weaving representations of MTS: $\mathbf{X}_{wa}$, pruning threshold for variable percentage: *Thr*

**Output:** Filtered variable pairs: $C$

    ▽ Step 1: Time lag discovery

1: **for** any variable pair $(v, v')$ **do**
2:     **for** any possible time lag $k \in [1, T - \omega + 1]$ **do**
3:         Compute $\mathbf{Xor}\langle v, v' \rangle$ for time lag $k$
4:         Calculate percentages of (0,0), (1,1), and (0,1)/(1,0) tuples: $R0[v, v', k]$, $R2[v, v', k]$, $R1[v, v', k]$
5:     **end for**
6: **end for**
7: $R[v, v', k] \leftarrow \max(R0[v, v', k], R2[v, v', k])$
8: Identify $(v, v')$ and $k$ with the highest value in $R[v, v', k]$, and store as $C \leftarrow \{(v, v'), k\}$
    ▽ Step 2: Variable candidates pruning
9: **for** each candidate $(v, v')$ in $C$ **do**
10:     **if** $R[v, v'] > 0.33$ **and** $|C| < Thr \cdot d^2$ **then**
11:         Retain $(v, v')$
12:     **else**
13:         Remove $(v, v')$ from $C$
14:     **end if**
15: **end for**

---

and $R1[v, v', k]$ (lines 1–5). For each time lag, the algorithm selects the maximum value between $R0[v, v', k]$ and $R2[v, v', k]$ as $R[v, v', k]$ (line 7). The variable pairs and their corresponding time lags with the highest $R[v, v', k]$ values are stored in the initial candidate set $C$ (line 8).

**Step 2: Variable Pairs Pruning.** We further filter the candidates in $C$. For each candidate pair $(v, v')$, it checks two conditions: (1) whether the value of $R[v, v']$ exceeds 0.33, and (2) whether the total number of candidates $|C|$ is below the threshold $Thr \cdot d^2$. If both conditions are met, the pair is retained in the final set; otherwise, it is removed (lines 10–14).

This algorithm analyzes time-lagged relationships in a multivariate time series to filter significant variable pairs. It computes XOR values to evaluate patterns and prunes results based on predefined thresholds, ensuring only the most relevant variable pairs are retained in the output.

## 5. Experiments

### 5.1. Experimental Setting

Datasets. We generate random summary graphs to simulate causal relationships using both linear and non-linear structural causal models (SCMs). For linear causal relationships, the SCM is defined as: $x_i^t = \sum_{x_j \in \mathbf{PA}(x_i)} \alpha_{ij} x_j (t - k_{ij}) + N_i$, where $\alpha_{ij}$ is a constant, $k_{ij}$ denotes the time lag, and $N_i$ is a noise selected from {*uniform*, *gauss*, *exp*, *gamma*}

randomly. For non-linear causal relationships, we adopt the Erd6s-Renyi model (Erd6s & Rényi, 1960; Liu et al., 2024) to generate the graph structure. Each non-linear SCM is defined as: $x_i^t = \sum_{x_j \in \mathbf{PA}(x_i)} f_{ij} x_j (t - k_{ij}) + N_i$, where $f_{ij}$ is randomly chosen from {*linear*, *sin*, *tanh*, *sqrt*}, $k_{ij}$ denotes time lag and $N_i$ is also noise sampled from the same distributions as in the linear case. In addition to the synthetic datasets, we also conduct validation on the real-world Dream3 [1] dataset.

Baselines. We select a representative algorithm from each of the four causal discovery approaches $M$ for our study: the constraint-based algorithm *PCMCI* (Runge et al., 2019; Runge, 2020), the Granger-based algorithm *NGC* (Tank et al., 2021), the score-based algorithm *VAR-LiNGAM* (Hyvärinen et al., 2010), and the information-theoretic algorithm *SURD* (Martínez-Sánchez et al., 2024). For each algorithm, we compare the performance of the original version with its accelerated counterpart using the *Arrow* accelerator, focusing on improvements in both efficiency and accuracy. Detailed descriptions of the methods can be found in Appendix C.1.

Metrics. We evaluate the efficiency of causal graph generation based on runtime (in seconds). Additionally, their time lag discovery and causal graph generation performance is assessed using three metrics: True Positive Rate (TPR), False Positive Rate (FPR), and Area Under the Curve (AUC). Details are shown in Appendix C.2.

Settings. We synthesized a dataset with 10 variables and a time length of 1000. The window size $w$ for time weaving was set to 1. In addition, we conducted experiments with constant time lags and multiple time lags. For the constant time lags, the lags between variables is fixed, and it can be selected from the set {3, 5, 7, 9, 15, 20}. In contrast, the multiple time lags represent varying lags between variables, with the lag value being chosen from the set {3, 5, 7, 9, 15, 20} as the range for the time lags. The experiments on varying time-lagged edges are deferred to Appendix D, while in the main experiments, we set $k$ to {5, 15}. The source code of ARROW is available [2].

### 5.2. Overall Performance

**Efficiency Evaluation.** For MTS data with linear causal relationships, as shown in Table 1, ARROW significantly improves the efficiency of causal discovery because its acceleration strategy efficiently reduces the search space for time lag and variable candidates, streamlining the computation process. *SURD* with ARROW achieves remarkable acceleration, with a nearly 70x speedup observed on the *SURD* dataset. This is primarily due to its ability to preemp-

---

[1] https://www.synapse.org/Synapse:syn3033083/files/
[2] https://github.com/XiangguanMu/arrow

*Table 1.* Performance on the datasets with linear casual relationships

| Constant Lags | | | | | | |
| --- | --- | --- | --- | --- | --- | --- |
| Time lag | 5 | | | 15 | | |
| Metric | **Time ↓** | **Lag AUC ↑** | **Graph AUC ↑** | **Time ↓** | **Lag AUC ↑** | **Graph AUC ↑** |
| PCMCI | 245.99±15.85 | 0.9350±0.053 | 0.9169±0.067 | 266.55±19.85 | 0.9275±0.031 | 0.9088±0.043 |
| PCMCI+ARROW | **6.6734±0.026** | **0.9775±0.018** | **0.9863±0.013** | **6.6969±0.021** | **0.9725±0.018** | **0.9906±0.010** |
| SURD | 424.90±1.837 | 0.5000±0.000 | 0.5038±0.008 | 422.59±0.677 | 0.5000±0.000 | 0.5200±0.017 |
| SURD+ARROW | **6.1656±0.013** | **0.9825±0.016** | **0.9644±0.013** | **6.1874±0.022** | **0.9700±0.042** | **0.9636±0.072** |
| NGC | 12.905±0.317 | 0.5000±0.000 | 0.7513±0.088 | 12.946±0.293 | 0.5000±0.000 | 0.8056±0.040 |
| NGC+ARROW | **0.7971±0.056** | **0.9750±0.027** | **0.9975±0.008** | **0.7931±0.066** | **0.9850±0.017** | **0.9906±0.018** |
| VARLiNGAM | 9.7244±0.439 | **0.9800±0.022** | 0.9606±0.021 | 9.3213±0.164 | 0.5000±0.000 | 0.4956±0.011 |
| VARLiNGAM+ARROW | **6.2020±0.013** | 0.9775±0.021 | **0.9819±0.014** | **6.2664±0.015** | **0.9850±0.023** | **0.9281±0.062** |
| Multiple Lags | | | | | | |
| Time lag | 5 | | | 15 | | |
| Metric | **Time ↓** | **Lag AUC ↑** | **Graph AUC ↑** | **Time ↓** | **Lag AUC ↑** | **Graph AUC ↑** |
| PCMCI | 388.41±83.10 | 0.7550±0.042 | 0.7381±0.040 | 441.16±55.09 | 0.6575±0.048 | 0.6369±0.067 |
| PCMCI+ARROW | **6.4486±0.031** | **0.8050±0.029** | **0.8088±0.030** | **6.4285±0.029** | **0.9125±0.023** | **0.9250±0.012** |
| SURD | 425.72±0.875 | 0.5050±0.010 | 0.5369±0.026 | 428.54±1.353 | 0.5000±0.000 | 0.5181±0.017 |
| SURD+ARROW | **6.1871±0.033** | **0.8925±0.016** | **0.9025±0.014** | **6.1711±0.024** | **0.9550±0.019** | **0.9394±0.023** |
| NGC | 13.043±0.339 | 0.5025±0.008 | 0.8263±0.046 | 13.021±0.278 | 0.5025±0.007 | 0.7688±0.031 |
| NGC+ARROW | **0.7482±0.014** | **0.8800±0.027** | **0.9469±0.036** | **0.7477±0.013** | **0.8775±0.028** | **0.9638±0.027** |
| VARLiNGAM | 9.1697±0.155 | **0.9925±0.016** | **0.9856±0.018** | 9.3054±0.129 | 0.6100±0.051 | 0.6100±0.051 |
| VARLiNGAM+ARROW | **6.2439±0.037** | 0.8900±0.025 | 0.8388±0.107 | **6.2743±0.044** | **0.8850±0.060** | **0.8294±0.135** |

tively filter and prioritize candidate lags, avoiding exhaustive searches that slow down the original algorithm. Additionally, ARROW's acceleration strategy ensures greater stability compared to the original algorithm because it applies the same optimization method for both constant lags and multiple lags settings, thereby minimizing variations in performance. *VARLiNGAM* outperforms other algorithms in terms of running time due to its Independent Component Analysis (ICA) strategy, but it still requires traversing the time lag range to find the optimal time lag. In contrast, *VARLiNGAM* with ARROW avoids the traversal to further accelerate the entire process. Note that, although ICA accelerates the casual discovery of *VARLiNGAM*, it is not a general optimization technique, which cannot be applied to other types of methods. Thus, our ARROW is a general accelerator that can significantly improve the efficiency of causal discovery on all the methods.

Similarly, when dealing with nonlinear causal relationships, ARROW's acceleration strategy still demonstrates significant efficiency as shown in Table 2, especially for *PCMCI*. This is because *PCMCI* uses partial correlation for CI test, which is more suited for linear causal relationships. When handling non-linear causal relationships, additional non-linear extensions are required, which significantly increase the computational cost.

**Effectiveness Evaluation in Time Lag Discovery.** As shown in Tables 1 and 2, ARROW consistently outperforms the original algorithm in terms of AUC for time lag discovery (i.e., Lag AUC) in most cases, when handling linear or nonlinear causal relationships across various lag sizes in both constant and multiple lags settings. This performance improvement is particularly noticeable on the *SURD* algorithm, where the maximum improvement reaches 46%. This is because: (i) as the number of variables increases, the conditional probability distribution computed by *SURD* tends to dilute the relevant variable information; (ii) the brute-force search method does not guarantee the discovery of the optimal time lag if "optimal" is not well defined. In addition, *VARLiNGAM* with *Arrow* performs worse than *VARLiNGAM* when the time lag is small, primarily because *VARLiNGAM* is suited for data with non-Gaussian noise, while the time weaving representation of *Arrow* may influence the original noise distribution, affecting the performance. However, when the time lag is larger, the robustness of causal relationships between variables is enhanced, significantly reducing the interference of the time weaving

*Table 2.* Performance on the datasets with non-linear casual relationships

**Constant Lags**

| Time lag | 5 | | | 15 | | |
|---|---|---|---|---|---|---|
| Metric | Time ↓ | Lag AUC ↑ | Graph AUC ↑ | Time ↓ | Lag AUC ↑ | Graph AUC ↑ |
| PCMCI | 1310.7±304.4 | 0.9156±0.064 | **0.9029±0.054** | 996.89±182.9 | 0.8772±0.056 | 0.8595±0.051 |
| PCMCI+Arrow | **9.6132±3.044** | **0.9342±0.035** | 0.8999±0.037 | **22.925±18.55** | **0.9167±0.046** | **0.8882±0.042** |
| SURD | 424.14±0.423 | 0.5000±0.000 | 0.5000±0.000 | 428.09±1.220 | 0.5000±0.000 | 0.4991±0.019 |
| SURD+ARROW | **6.3629±0.059** | **0.9470±0.039** | **0.7310±0.109** | **6.3844±0.145** | **0.9315±0.048** | **0.7642±0.060** |
| NGC | 12.980±0.318 | 0.5000±0.000 | 0.4391±0.003 | 12.960±0.293 | 0.5000±0.000 | **0.4406±0.001** |
| NGC+ARROW | **0.7430±0.013** | **0.9563±0.036** | **0.4485±0.033** | **0.7415±0.013** | **0.9131±0.047** | 0.3889±0.075 |
| VARLiNGAM | 9.0475±0.171 | **0.9944±0.011** | **0.9834±0.013** | 9.0925±0.104 | 0.5000±0.000 | 0.4970±0.013 |
| VARLiNGAM+ARROW | **6.2198±0.021** | 0.9133±0.057 | 0.8974±0.020 | **6.2424±0.021** | **0.9352±0.053** | **0.9057±0.022** |

**Multiple Lags**

| Time lag | 5 | | | 15 | | |
|---|---|---|---|---|---|---|
| Metric | Time ↓ | Lag AUC ↑ | Graph AUC ↑ | Time ↓ | Lag AUC ↑ | Graph AUC ↑ |
| PCMCI | 1076.7±196.8 | 0.655±0.068 | 0.6651±0.075 | 1368.8±222.4 | 0.6575±0.048 | 0.6368±0.067 |
| PCMCI+ARROW | **7.6967±1.387** | **0.8774±0.058** | **0.8344±0.059** | **16.591±13.15** | **0.9157±0.063** | **0.8680±0.066** |
| SURD | 426.65±1.340 | 0.5063±0.013 | 0.4915±0.013 | 423.34±0.141 | 0.5000±0.000 | 0.4840±0.010 |
| SURD+ARROW | **6.1981±0.032** | **0.8836±0.030** | **0.7108±0.046** | **6.2153±0.036** | **0.9477±0.049** | **0.6877±0.060** |
| NGC | 13.155±0.340 | 0.5000±0.000 | 0.4379±0.002 | 12.906±0.304 | 0.5000±0.000 | **0.4398±0.003** |
| NGC+ARROW | **0.7410±0.013** | **0.8916±0.090** | **0.4382±0.004** | **0.7478±0.014** | **0.8983±0.033** | 0.4060±0.017 |
| VARLiNGAM | 9.0378±0.129 | **0.9292±0.077** | **0.9198±0.073** | 9.0330±0.191 | 0.5549±0.054 | 0.5609±0.052 |
| VARLiNGAM+ARROW | **6.2423±0.014** | 0.8135±0.047 | 0.9064±0.038 | **6.2934±0.026** | **0.8931±0.044** | **0.9122±0.023** |

representation on the original distribution.

**Effectiveness Evaluation in Summary Graph Generation.** Similar to the performance in time lag discovery, the AUC for summary graph generation (i.e., Graph AUC) also outperforms the original algorithm, as shown in Tables 1 and 2. This improvement benefits from ARROW's strategies in time lag discovery and candidate pruning. In addition, the improvement is consistent across datasets with both constant lags and multiple lags settings, handling both linear and nonlinear causal relationships, demonstrating ARROW's robustness and efficiency in generating reliable causal graphs.

### 5.3. Experiments on the Real-world Dataset

We validate the acceleration performance of the ARROW framework on the real-world time series dataset Dream3. The results as shown in Figure 4 demonstrate that ARROW achieves 7x to 300x speedup across various methods, with particularly remarkable performance on the *SURD* method. This is because the *SURD* method relies on a brute-force approach for time lag selection, whereas ARROW significantly improves search efficiency and accuracy through its optimized time lag mining algorithm and pruning strategy.

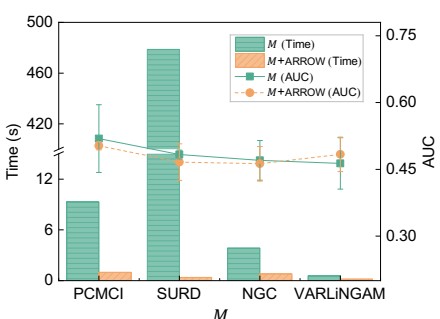

*Figure 4.* Causal discovery performance on Dream3

Furthermore, in terms of summary graph performance, ARROW successfully retains the performance of the original algorithms and even achieves better results on the *VARLiNGAM* method. This is because *VARLiNGAM* is suitable for datasets with non-Gaussian noise while the Dream3 dataset contains heterogeneous noise components.

### 6. Conclusion

We propose a general accelerator for time series causal discovery algorithms termed ARROW. First, we introduce a novel concept called "Time Weaving", which transforms

data from a single time point into a binary tuple with contextual information. Next, we propose a time lag discovery strategy that analyzes contextual trend patterns between different variables. We derive a theorem to find the optimal time lag and provide rigorous proofs. Additionally, we develop a variable candidates pruning algorithm to effectively reduce the candidate set for causal discovery, further improving computational efficiency. Finally, experiments on both synthetic and real datasets demonstrate that ARROW can significantly accelerate various causal discovery algorithms while maintaining high accuracy.

## Acknowledgement

This work was supported in part by the NSFC under Grants No. (62472377, 62402422). Lu Chen is the corresponding author of the work.

## Impact Statement

This paper presents work whose goal is to advance the field of causality. There are many potential societal consequences of our work, none which we feel must be specifically highlighted here.

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

# A. Proofs

**Theorem A.1.** *Given two variables $v$ and $v'$ in a MTS data $\mathbf{X}$, if $v$ and $v'$ have a causal relationship in a time lag $k$, then either $P(\mathbf{Xor}(v, v', k) = (0, 0))$ or $P(\mathbf{Xor}(v, v', k) = (1, 1))$ will exceed $\frac{1}{3} - \epsilon$, where $\epsilon$ is a value approaching 0.*

*Proof.* Let $\Gamma = \{(0, 0), (0, 1), (1, 0), (1, 1)\}$ represent the possible tuple set resulting from the XOR operation between time weaving representation of $v$ and $v'$ (i.e., $\mathbf{Xor}\langle v, v'\rangle$) at time point $t$ with time lag $k$. For any tuple $\tau \in \Gamma$, there is a corresponding tuple $\tau^c$ that represents a completely opposite trend (e.g., (1,0)-(0,1), (0,0)-(1,1)). Then, there are three different combination results:

$$P(\mathbf{Xor}(v, v', k) = (0, 0)) = \sum_{\tau \in \Gamma} P(x_{v'}^t = \tau, x_v^{t-k} = \tau)$$

$$P(\mathbf{Xor}(v, v', k) = (1, 1)) = \sum_{\tau \in \Gamma} P(x_{v'}^t = \tau^c, x_v^{t-k} = \tau) \tag{2}$$

$$P(\mathbf{Xor}(v, v', k) \in \{(0, 1), (1, 0)\}) = \sum_{\tau \in \Gamma} P(x_{v'}^t \in \Gamma - \{\tau, \tau^c\}, x_v^{t-k} = \tau)$$

where $P(\mathbf{Xor}(v, v', k) = (0, 0))$ represents the probability that the contexts of $x_{v'}^t$ and $x_v^{t-k}$ exhibit a similar trend, while $P(\mathbf{Xor}(v, v', k) = (1, 1))$ represents the probability of an opposite trend. $P(\mathbf{Xor}(v, v', k) \in \{(0, 1), (1, 0)\})$ represents the probability that the contexts of $x_{v'}^t$ and $x_v^{t-k}$ share some similarities or differs in others. The joint distribution $P(x_{v'}^t, x_v^{t-k})$ of two variables $v$ and $v'$ can be factored using the conditional probability formula as:

$$P(x_{v'}^t = \tau, x_v^{t-k} = \tau) = P(x_{v'}^t = \tau | x_v^{t-k} = \tau)P(x_v^{t-k} = \tau)$$

$$P(x_{v'}^t = \tau^c, x_v^{t-k} = \tau) = P(x_{v'}^t = \tau^c | x_v^{t-k} = \tau)P(x_v^{t-k} = \tau) \tag{3}$$

$$P(x_{v'}^t \in \Gamma - \{\tau, \tau^c\}, x_v^{t-k} = \tau) = P(x_{v'}^t \in \Gamma - \{\tau, \tau^c\} | x_v^{t-k} = \tau)P(x_v^{t-k} = \tau)$$

Here, $P(x_v^{t-k})$ is the marginal distribution of $v$, and $P(x_{v'}^t \mid x_v^{t-k})$ is the conditional distribution of $v'$ given the state $x_v^{t-k}$ of $v$.

Given $x_v^{t-k} = \tau$, if $v$ has a positive causal effect on $v'$, then the probability of $x_{v'}^t = \tau$ tends to be higher than that of $x_{v'}^t = \tau'$, where $\tau' \in \Gamma - \{\tau, \tau^c\}$, as illustrated in Equation 4. Furthermore, the more pronounced the positive causal effect, the closer the probability of indicating an opposite trend, $P(x_{v'}^t = \tau^c | x_v^{t-k} = \tau)$, approaches zero.

$$P(x_{v'}^t = \tau | x_v^{t-k} = \tau) > P(x_{v'}^t = \tau' | x_v^{t-k} = \tau), \quad \tau' \in \Gamma - \{\tau, \tau^c\} \tag{4}$$

$$0 < P(x_{v'}^t = \tau^c | x_v^{t-k} = \tau) < \epsilon_0 \tag{5}$$

where $\epsilon_0$ represents a probability approaching 0. Note that $P(x_{v'}^t = \tau' | x_v^{t-k} = \tau)$ could not be zero because it may be potentially influenced by other external factors, such as noise or other variables in SCM.

Therefore, given $x_v^{t-k} = \tau$, the sum of the probabilities for all possible contextual trends of $x_{v'}^t$ ( the similar trend, opposite trend, or other cases) must equals to 1 as shown in Equation 6. Based on Equations 5 and 6, we can further derive Equation 7.

$$P(x_{v'}^t = \tau | x_v^{t-k} = \tau) + \sum_{\tau' \in \Gamma - \{\tau, \tau^c\}} P(x_{v'}^t = \tau' | x_v^{t-k} = \tau) + P(x_{v'}^t = \tau^c | x_v^{t-k} = \tau) = 1 \tag{6}$$

$$P(x_{v'}^t = \tau | x_v^{t-k} = \tau) + \sum_{\tau_i \in \Gamma - \{\tau, \tau^c\}} P(x_{v'}^t = \tau_i | x_v^{t-k} = \tau) > 1 - \epsilon_0 \tag{7}$$

Given Equation 4, we can infer that given $x_v^{t-k} = \tau$, the conditional probability of $x_{v'}^t = \tau$ is greater than the average of the

probabilities for all other possible trends as indicated in Equation 8.

$$\sum_{|\Gamma|-1} P(x_{v'}^t = \tau | x_v^{t-k} = \tau) > P(x_{v'}^t = \tau | x_v^t = \tau) + \underbrace{P(x_{v'}^t = \tau_1' | x_v^t = \tau) + \cdots P(x_{v'}^t = \tau_{|\Gamma-\{\tau,\tau^c\}|}' | x_v^{t-k} = \tau)}_{|\Gamma|-2}$$

$$P(x_{v'}^t = \tau | x_v^{t-k} = \tau) > \frac{1}{|\Gamma|-1}(P(x_{v'}^t = \tau | x_v^{t-k} = \tau) + \sum_{\tau' \in \Gamma - \{\tau,\tau^c\}} P(x_{v'}^t = \tau' | x_v^{t-k} = \tau))$$

$$> \frac{1}{|\Gamma|-1}(1 - \epsilon_0)$$

$$= \frac{1}{3} - \epsilon, \quad \epsilon = \frac{1}{3}\epsilon_0$$

(8)

Due to $\sum_{\tau \in \Gamma} P(x_v^{t-k} = \tau) = 1$, we can infer the Equation 9.

$$P(\mathbf{Xor}(v, v', k) = (0,0)) = \sum_{\tau \in \Gamma} P(x_{v'}^t = \tau, x_v^{t-k} = \tau)$$

$$= \sum_{\tau \in \Gamma} P(x_{v'}^t = \tau | x_v^{t-k} = \tau) P(x_v^{t-k} = \tau)$$

$$> (\frac{1}{3} - \epsilon) \sum_{\tau \in \Gamma} P(x_v^{t-k} = \tau)$$

$$= \frac{1}{3} - \epsilon$$

(9)

In general, if $v$ has a positive causal effect on $v'$, $P(\mathbf{Xor}(v, v', k) = (0,0))$ should be greater than $\frac{1}{3} - \epsilon$. Similarly, if $v$ has a negative causal effect on $v'$, $P(\mathbf{Xor}(v, v', k) = (1,1))$ should be greater than $\frac{1}{3} - \epsilon$. It also proves why the threshold selected (i.e., 0.33) in the Algorithm 1 is slightly smaller than $\frac{1}{3}$.

$\square$

# B. Discussions

Our acceleration framework is compatible with various causal discovery algorithms, supports irregular time series data input, and enables efficient concurrent causal discovery.

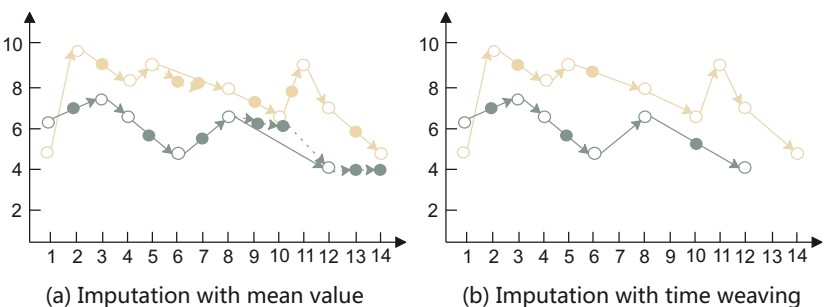

(a) Imputation with mean value    (b) Imputation with time weaving

*Figure 5.* Imputation for irregular MTS

**Adaptive to irregular MTS.** Most causal discovery algorithms typically handle irregular time series by imputing the missing values using imputation methods (Liu et al., 2024; Cheng et al., 2024; 2023), such as imputation by mean value as illustrated in Figure 5(a). However, this approach may alter the original distribution of the data, especially when too many consecutive missing values exists. In contrast, ARROW adopts time-weaving representations instead of imputation, effectively preserving the original data distribution, as illustrated in Figure 5(b). For any two variables $v$ and $v'$ in irregular multivariate time series (MTS) data $\mathbf{X}$, $v$ contains 6 time points $\{1, 3, 4, 6, 8, 12\}$, while $v'$ contains 8 different time points $\{1, 2, 4, 5, 8, 10, 11, 14\}$. ARROW constructs the union of their time points, $T < v, v' >= T_v \cup T_{v'}$, resulting in a complete time set $\{1, 2, 3, 4, 5, 6, 8, 10, 11, 12, 14\}$.

Next, the values of the variables are transformed using the time-weaving transformation. For any time point in the union set $(T < v, v' > -T_v$ or $T < v, v' > -T_{v'})$ where a variable has missing values, the missing values are filled based on the trend between adjacent time points. For example, for variable $v$, the missing value at $t = 5$ is filled using the trend between $t = 4$ and $t = 6$. Similarly, for variable $v'$, the missing values at $t = 6$ and $t = 7$ are filled using the trend between $t = 5$ and $t = 8$.

Compared to traditional imputation methods, ARROW preserves the original trends and distribution of the data through time-weaving representations, avoiding the bias introduced by imputation assumptions. It naturally handles data with irregular time points and better adapts to the complexity of multivariate time series data.

**Adaptive to parallel acceleration.** In causal discovery algorithms, CUDA-based parallel computation strategies can significantly enhance computational efficiency, particularly when exploring causal relationships between variables. CUDA leverages the parallel processing capabilities of GPUs to accelerate logical operations such as AND, OR, and NOT, which are commonly applied to 0/1 data. The inherent simplicity and low memory consumption of 0/1 data make it well-suited for parallel computation, enabling CUDA to process large-scale datasets efficiently and dramatically reduce computation time, thereby improving scalability and real-time performance.

Specifically, in the case of ARROW, the original data is discretized into 0 and 1 values, making it particularly compatible with CUDA acceleration. This discretization reduces memory and bandwidth overhead, while the simplicity of logical operations on binary data allows CUDA to perform parallel computations more effectively. As a result, ARROW benefits from both the data transformation and parallel processing, providing an efficient solution for causal discovery tasks, especially in large-scale and complex time series data analysis. This combination of data discretization and CUDA parallelism significantly enhances the computational performance and processing speed of causal discovery algorithms.

## C. Experimental Setting

### C.1. Baselines description

In our experiments, we selected four representative algorithms for detailed study: the constraint-based algorithm *PCMCI*, the information-theoretic algorithm *SURD*, the Granger-based algorithm *NGC*, and the score-based algorithm *VARLiNGAM*.

- *PCMCI* is a constraint-based causal discovery algorithm that improves the original PC algorithm for application to time series data, aiming to uncover causal relationships between variables in time series data. The algorithm combines conditional independence tests with machine learning methods. The key idea behind *PCMCI* is to use "post-effects" (Post-Nonlinear) in time series data to capture potential nonlinear causal relationships. In the experiment, we implemented the *PCMCI* algorithm based on the open-source code [3]. *PCMCI* can generate causal graphs for all candidate time lags, and we determine the time lag and final causal graph by selecting the densest time lag set.
- *SURD* is an information-theoretic causal discovery method that uses the principle of uncertainty reduction in information theory to infer causal relationships. The algorithm infers the causal relationships between variables by calculating the information gain between them. *SURD* focuses on discovering causal relationships by reducing the system's entropy (i.e., uncertainty), making it effective in handling nonlinear and multivariable dependencies in complex systems. We implemented the *SURD* algorithm based on the open-source code [4] and implemented CUDA acceleration. We determine the optimal time lag and summary graph using the brute-force search method usned in the *SURD*.
- *NGC* is an algorithm based on granger causality, specifically designed for causal discovery in time series data. The Granger causality test assumes that if variable $X$ has a causal relationship with variable $Y$, then past values of $X$ can help predict future values of $Y$. The *NGC* algorithm extends the traditional granger causality test to capture nonlinear causal relationships between variables. This is achieved by performing granger causality testing on component-wise MLP model, thereby enhancing the ability to model complex dynamic systems. We implemented the *NGC* algorithm based on the open-source code [5]. *NGC* can directly return the time lag and summary graph.
- *VARLiNGAM* is a score-based causal discovery algorithm that combines the Vector Autoregressive (VAR) model with the Linear Non-Gaussian Acyclic Model (LiNGAM). This method uses the VAR model to describe the temporal dependencies between variables and performs causal discovery based on the linear non-Gaussian assumption. *VARLiNGAM* assumes that the error terms of the variables follow a non-Gaussian distribution, enabling it to distinguish between causal relationships.

---

[3]https://github.com/jakobrunge/tigramite
[4]https://github.com/Computational-Turbulence-Group/SURD
[5]https://github.com/shojaie/ngc

We implemented the *VARLiNGAM* algorithm based on the open-source code [6]. *VARLiNGAM* outputs a set of candidate causal graphs, and we obtain the final causal graph by averaging these candidates. Based on the contribution of each candidate graph, we use the time lag with the highest contribution as the optimal time lag for each position.

## C.2. Metric details

In our experiments, we not only evaluated the efficiency of the algorithms but also assessed the effectiveness of causal discovery, including the performance of summary graph generation and time lag discovery.

To evaluate the performance of different algorithms in generating causal graphs, the following metrics are commonly used. Let the inferred edge probability from the algorithm be $P(A_{ij})$, with a threshold $thre \in (0, 1)$. The set of edges in the ground truth causal graph is $\mathbf{E}_T$, and the set of missing edges is $\mathbf{E}_M$. The definitions of metrics are as follows:

$$
\begin{aligned}
TPR_G &= \frac{|\{(i,j) : P(A_{ij}) \geq thre\} \cap E_T|}{|E_T|}, \\
FPR_G &= \frac{|\{(i,j) : P(A_{ij}) \geq thre\} \cap E_S|}{|E_S|}.
\end{aligned}
\tag{10}
$$

TPR (True Positive Rate) represents the proportion of correctly identified edges to the total number of true edges, serving as a measure of the algorithm's ability to recognize true edges—the higher, the better. FPR (False Positive Rate) indicates the proportion of incorrectly identified edges to the total number of missing edges, reflecting the algorithm's tendency to misjudge non-existent edges—the lower, the better. The ROC curve is constructed by plotting the relationship between TPR and FPR, while AUROC (Area Under the ROC Curve) measures the area under this curve, i.e., $AUC = \sum_{i=1}^{n-1} \frac{(FPR_{i+1} - FPR_i) \cdot (TPR_{i+1} + TPR_i)}{2}$. It is a key metric for assessing the overall performance of an algorithm. The closer the AUROC value is to 1, the better the algorithm performs.

To evaluate the performance of time lag discovery, we also use the three metrics TPR, FPR, and AUC. Let $P(A_{ij}, k)$ represent the probability of an edge between $i$ and $j$ at time lag $k$, with the maximum time lag denoted as Max. The time lag in the true edges is represented by $\mathbf{TL}_T$, while the time lag in the missing edges is represented by $\mathbf{TL}_M$. The definitions of the time lag discovery metrics are as follows:

$$
\begin{aligned}
TPR_{TL} &= \frac{\left|\{(i,j) : \max_{k \in [1,\text{Max}]} P(A_{ij}, k) > 0.33\} \cap TL_T\right|}{|TL_T|}, \\
FPR_{TL} &= \frac{\left|\{(i,j) : \max_{k \in [1,\text{Max}]} P(A_{ij}, k) > 0.33\} \cap TL_S\right|}{|TL_S|}.
\end{aligned}
\tag{11}
$$

where $\max_{k \in [1,\text{Max}]} P(A_{ij}, k)$ indicates iterating over all $k$ values (from 1 to Max) and selecting the maximum probability for each $(i, j)$ pair. $P(A_{ij}, k) > 0.33$ signifies selecting edges where the maximum probability value for the corresponding $(i, j)$ pair exceeds 0.33. TPR represents the proportion of edges with actual delays that are correctly detected, while FPR represents the proportion of edges without delays that are incorrectly identified as having delays.

## C.3. Experimental environment

All methods are executed on a machine equipped with an Intel(R) Core(TM) i9-10900K CPU, boasting 10 cores and a clock speed of 3.70GHz. The system also features an NVIDIA GeForce RTX 3090 graphics card, equipped with 24GB of video memory.

# D. Extensive Experiments

## D.1. Efficiency Evaluation

We do extensive experiments on the four algorithms with $k$ selected from the set of $\{3, 5, 7, 9, 15, 20\}$. The recorded time includes both time lag detection and causal graph discovery time, except for *NGC*, which is a learning method using training

---

[6]https://causal-learn.readthedocs.io/en/latest/search_methods_index/Causal

*Table 3.* Comparison on causal discovery time

| | | | Linear+Constant Lags | | | |
|---|---|---|---|---|---|---|
| Time lag | 3 | 5 | 7 | 9 | 15 | 20 |
| PCMCI | 240.31±19.47 | 245.99±15.85 | 244.87±19.75 | 258.45±23.13 | 266.55±19.85 | 288.10±24.52 |
| PCMCI+Arrow | **6.6506±0.031** | **6.6734±0.026** | **6.6722±0.021** | **6.6966±0.026** | **6.6969±0.021** | **6.6868±0.021** |
| SURD | 424.00±2.001 | 424.90±1.837 | 425.34±2.040 | 423.98±1.228 | 422.59±0.677 | 421.91±0.228 |
| SURD+Arrow | **6.1697±0.026** | **6.1656±0.013** | **6.1681±0.011** | **6.1803±0.016** | **6.1874±0.022** | **6.1687±0.015** |
| NGC | 12.918±0.324 | 12.905±0.317 | 12.844±0.273 | 12.889±8.293 | 12.946±0.293 | 12.898±0.295 |
| NGC+Arrow | **0.7608±0.045** | **0.7971±0.056** | **0.7949±0.055** | **0.7952±0.058** | **0.7931±0.066** | **0.7484±0.013** |
| VARLiNGAM | 9.5531±0.660 | 9.7244±0.439 | 9.2401±0.137 | 9.2190±0.129 | 9.3213±0.164 | 9.5007±0.319 |
| VARLiNGAM+Arrow | **6.2040±0.013** | **6.2020±0.013** | **6.2192±0.020** | **6.2305±0.016** | **6.2664±0.015** | **6.2882±0.039** |
| | | | Linear+Multiple Lags | | | |
| Time lag | 3 | 5 | 7 | 9 | 15 | 20 |
| PCMCI | 306.21±38.92 | 388.41±83.10 | 416.26±36.13 | 442.87±85.53 | 441.16±55.09 | 491.77±95.42 |
| PCMCI+Arrow | **6.6167±0.540** | **6.4486±0.031** | **6.4982±0.026** | **7.2986±2.375** | **6.4285±0.029** | **6.4778±0.034** |
| SURD | 427.92±2.212 | 425.72±0.875 | 427.58±0.336 | 428.81±0.336 | 428.54±1.353 | 425.71±0.931 |
| SURD+Arrow | **6.2240±0.120** | **6.1871±0.033** | **6.1442±0.016** | **6.1598±0.014** | **6.1711±0.024** | **6.1761±0.034** |
| NGC | 15.108±4.025 | 13.043±0.339 | 13.010±0.273 | 13.059±0.282 | 13.021±0.278 | 13.039±0.273 |
| NGC+Arrow | **0.7432±0.013** | **0.7482±0.014** | **0.7473±0.014** | **0.7466±0.013** | **0.7477±0.013** | **0.7462±0.013** |
| VARLiNGAM | 8.8738±0.104 | 9.1697±0.155 | 9.2141±0.133 | 9.2228±0.110 | 9.3054±0.129 | 9.2058±0.126 |
| VARLiNGAM+Arrow | **6.2325±0.025** | **6.2439±0.037** | **6.2526±0.037** | **6.2522±0.420** | **6.2743±0.044** | **6.2668±0.026** |
| | | | Non-Linear+Constant Lags | | | |
| Time lag | 3 | 5 | 7 | 9 | 15 | 20 |
| PCMCI | 1160.8±287.2 | 1310.7±304.4 | 1386.1±384.8 | 1014.72±231.6 | 996.89±182.9 | 1287.8±264.2 |
| PCMCI+Arrow | **7.5468±1.346** | **9.6132±3.044** | **13.056±5.052** | **28.265±18.342** | **22.925±18.55** | **65.774±57.08** |
| SURD | 424.08±0.478 | 424.14±0.423 | 424.36±0.256 | 424.15±0.194 | 428.09±1.220 | 428.94±0.365 |
| SURD+Arrow | **6.4680±0.324** | **6.3629±0.059** | **6.3822±0.071** | **6.3667±0.028** | **6.3844±0.145** | **6.4870±0.129** |
| NGC | 12.895±0.281 | 12.980±0.318 | 12.855±0.308 | 12.899±0.290 | 12.960±0.293 | 12.832±0.292 |
| NGC+Arrow | **0.7406±0.015** | **0.7430±0.013** | **0.7390±0.013** | **0.7406±0.013** | **0.7415±0.013** | **0.7328±0.057** |
| VARLiNGAM | 9.1115±0.132 | 9.0475±0.171 | 9.0531±0.102 | 9.0653±0.104 | 9.0925±0.104 | 9.0630±0.095 |
| VARLiNGAM+Arrow | **6.2116±0.013** | **6.2198±0.021** | **6.2297±0.012** | **6.2448±0.024** | **6.2424±0.021** | **6.2322±0.017** |
| | | | Non-Linear+Multiple Lags | | | |
| Time lag | 3 | 5 | 7 | 9 | 15 | 20 |
| PCMCI | 1139.3±277.2 | 1076.7±196.8 | 1175.6±357.1 | 1185.7±262.1 | 1368.8±222.4 | 1033.2±263.3 |
| PCMCI+Arrow | **7.6503±1.325** | **7.6967±1.387** | **9.2900±4.339** | **10.7270±4.502** | **16.591±13.154** | **29.738±19.92** |
| SURD | 428.54±0.180 | 426.65±1.340 | 425.38±1.425 | 423.40±0.133 | 423.34±0.141 | 425.50±0.440 |
| SURD+Arrow | **6.2758±0.087** | **6.1981±0.032** | **6.2051±0.039** | **6.1967±0.039** | **6.2153±0.036** | **6.1909±0.028** |
| NGC | 13.246±0.326 | 13.155±0.340 | 12.973±0.275 | 12.964±0.312 | 12.906±0.304 | 12.985±0.306 |
| NGC+Arrow | **0.7390±0.016** | **0.7410±0.013** | **0.7439±0.013** | **0.7462±0.011** | **0.7478±0.014** | **0.7482±0.014** |
| VARLiNGAM | 8.8014±0.196 | 9.0378±0.129 | 8.9910±0.136 | 9.0197±0.137 | 9.0330±0.191 | 9.1033±0.193 |
| VARLiNGAM+Arrow | **6.2102±0.018** | **6.2423±0.014** | **6.2769±0.019** | **6.2796±0.016** | **6.2934±0.026** | **6.2852±0.016** |

time for each 100 epochs. Our method achieves up to 153x, 69x, 20x, and 1.57x speedup compared with *PCMCI*, *SURD*, *NGC*, and *VARLiNGAM* among synthetic datasets with linear and non-linear causal relationships as shown in Table 3, demonstrating a favorable accelerating effect.

Specifically, *PCMCI* with Arrow performs more stably across both linear and non-linear datasets, while *PCMCI* is significantly slower on non-linear datasets than linear ones. This is because CI tests on non-linear dependency between variables use more complicated calculations. Equipped with Arrow, MTS data is discretized into a binary representation, simplifying the CI tests and thus achieving a swift and stable performance.

*SURD* takes the longest time compared to other algorithms, and its time increases rapidly as the time lag range expands. This is because *SURD* uses a brute-force traversal method to detect the optimal time lag and calculates the joint and conditional

distributions for each variable and its candidate set. However, *SURD* with Arrow maintains approximately 6 seconds regardless of the time lag value. This is due to Arrow's pruning strategy, which significantly reduces the candidate set for each variable, exponentially decreasing the number of conditional distribution calculations.

*NGC* with Arrow spends less time per 100 epochs compared to the *NGC*. This is because the size of convolution kernel is set according to the time lag size. *NGC* requires the max time lag range, whereas Arrow only to set it to $k$ identified by the time lag discovery strategy, which is much smaller than the max value.

*VARLiNGAM* outperforms other algorithms in terms of efficiency due to its ICA, linear models, and recursive regression. However, it still requires traversing the time lag range to find the optimal lag. In contrast, *VARLiNGAM* with Arrow avoids the traversing, further speeding up the discovery. Note that, although ICA and other optimization techniques accelerate the casual discovery of *VARLiNGAM*, they are not general, and cannot be applied to other types of methods. To sum up, our Arrow is a general accelerator that can significantly improve the efficiency of causal discovery on all the methods.

### D.2. Effectiveness Evaluation in Time Lag Discovery

As shown in the Figures 6 and 7, the time lag discovery strategy of Arrow outperforms almost all the baselines.

*PCMCI* performs better on the dataset with constant lags than those with multiple lags. This is because *PCMCI* returns summary graphs for each possible time lag, and does not provide an internal implementation of selecting the best time lag for each variable. In addition, *PCMCI* with Arrow is more stable than *PCMCI* and performs better on the dataset with multiple lags. This is because the time lag discovery strategy of Arrow can adapt to datasets with different types of relationships.

*SURD* exhibits performance similar to that of a random classifier on both constant and multiple lags datasets, while *SURD* with Arrow is able to select a much more accurate lag graph. This is because: i) *SURD* assumes the lags among variables be equal, which makes it fail to find the optimal lag graph on datasets with multiple lags; ii) the real causation can be diluted when the size of candidate sets becomes too large. However, Arrow can select the most appropriate time lag for each variable pair in the candidate set by time lag discovery and candidate pruning strategies.

*NGC* with Arrow can achieve better time lag discovery performance than *NGC* on all datasets, and *VARLiNGAM* with Arrow is better than *VARLiNGAM* when the time lag exceeds 10. This is because both *NGC* and *VARLiNGAM* deduce the optimal lag graph by selecting the lag that contributes the most to the final summary graph, which ignores the contextual information in the original input data. In contrast, Arrow improves time lag discovery accuracy by preserving the contextual information of each data point through time weaving. In addition, the performance of *VARLiNGAM* slips sharply as the time lag exceeds 10, due to the non-gaussianity of the data distribution being obscured in cases of long lags.

### D.3. Effectiveness Evaluation in Summary Graph Generation

As shown in Figures 8 and 9, the accuracy of the summary graph generated by the original algorithm equipped with Arrow is much better than that of the original method on most datasets.

*PCMCI* with Arrow performs better than *PCMCI*, especially on the dataset with multiple lags. This is because *PCMCI* does not support multi-lag causation discovery, thus encountering a performance drop from constant-lagged to multiple-lagged datasets, while Arrow can effectively handle causality relationships with multiple lags.

*SURD* has a worse performance on deducing the summary causal graph because that is constructed on the wrongly-inferred lag graph. Moreover, the pruning strategy imported by However, if given the correct lag graph and pruned parent set inferred by Arrow, *SURD* can achieve a performance burst of at most 94%. This is because Arrow concentrates the conditional distribution on the potentially important subset of variables out of the entire, focusing the algorithm on more significant causal relationships for each pair of variables.

Both *NGC* and *NGC* with Arrow exhibit poor performance on non-linear datasets. This is because the items on the leading diagonal tends to be kept during the simplification of the initially fully connected causal graph. However, our synthetic Erdos-Renyi graph would not have its diagonal element to be 1, leading to the poor performance on our non-linear datasets.

*VARLiNGAM* with Arrow exhibits comparable excellence with *VARLiNGAM* when the time lag is small. However, their performance significantly declines as the lag increases, exhibiting a similar pattern and underlying reasons observed in the time lag discovery.

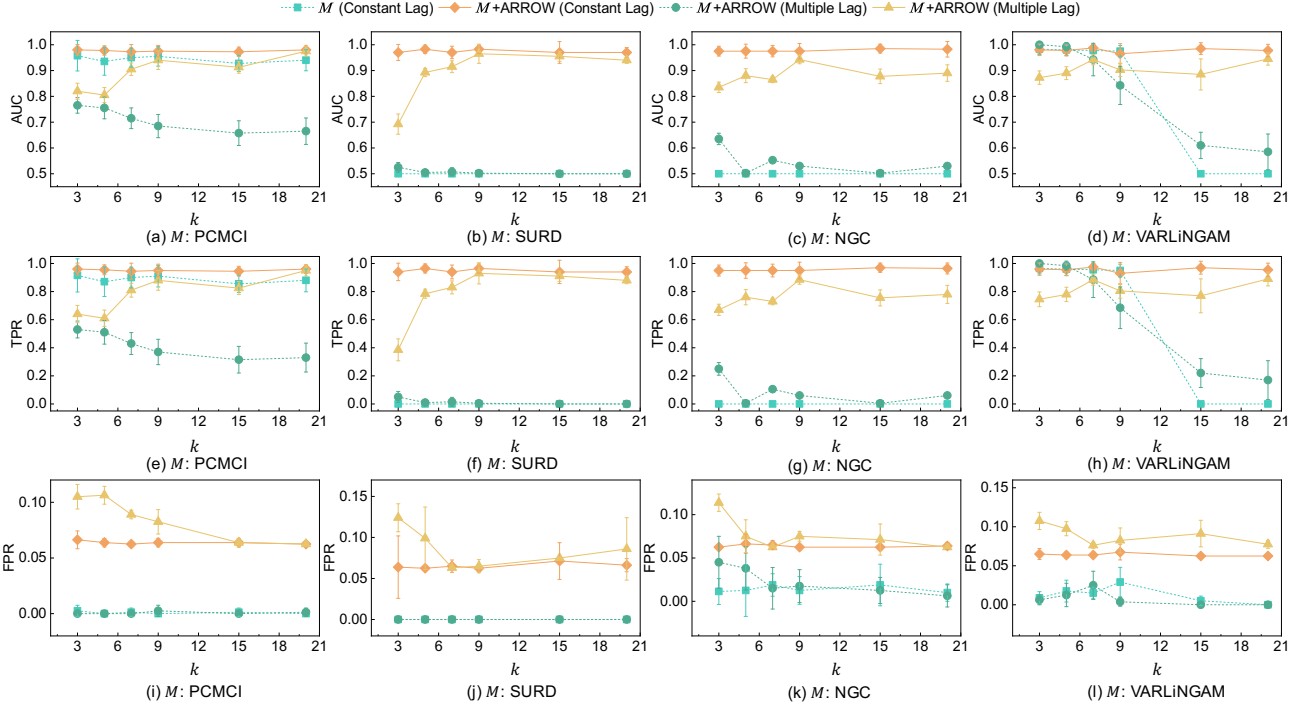

*Figure 6.* Time lag discovery performance on the dataset with linear causal relationships

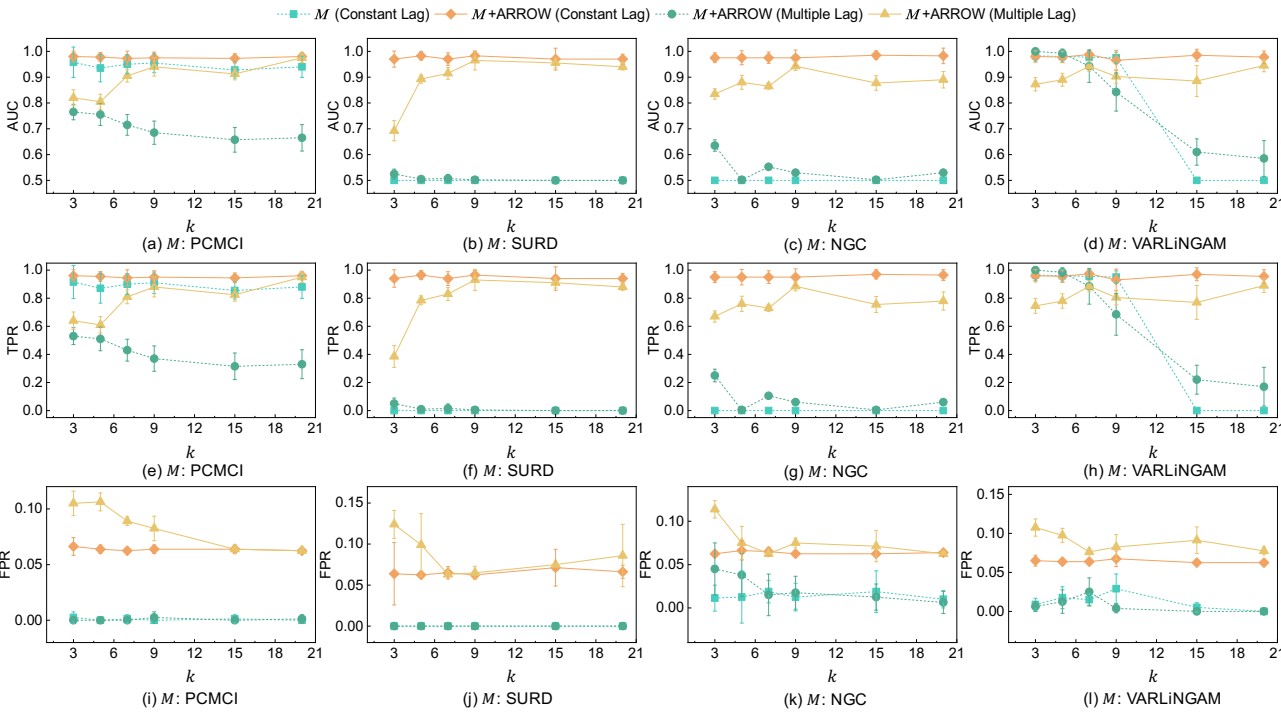

*Figure 7.* Time lag discovery performance on the dataset with non-linear causal relationships

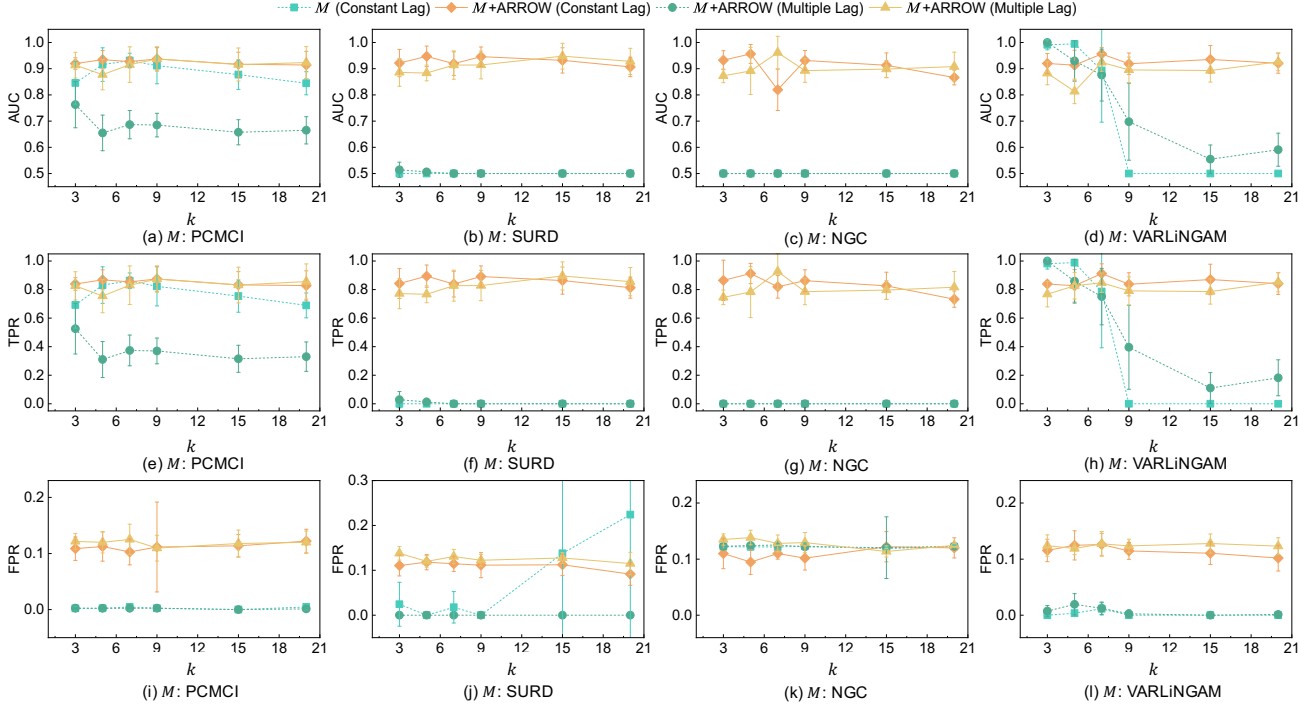

*Figure 8.* Summary graph performance on the dataset with linear causal relationships

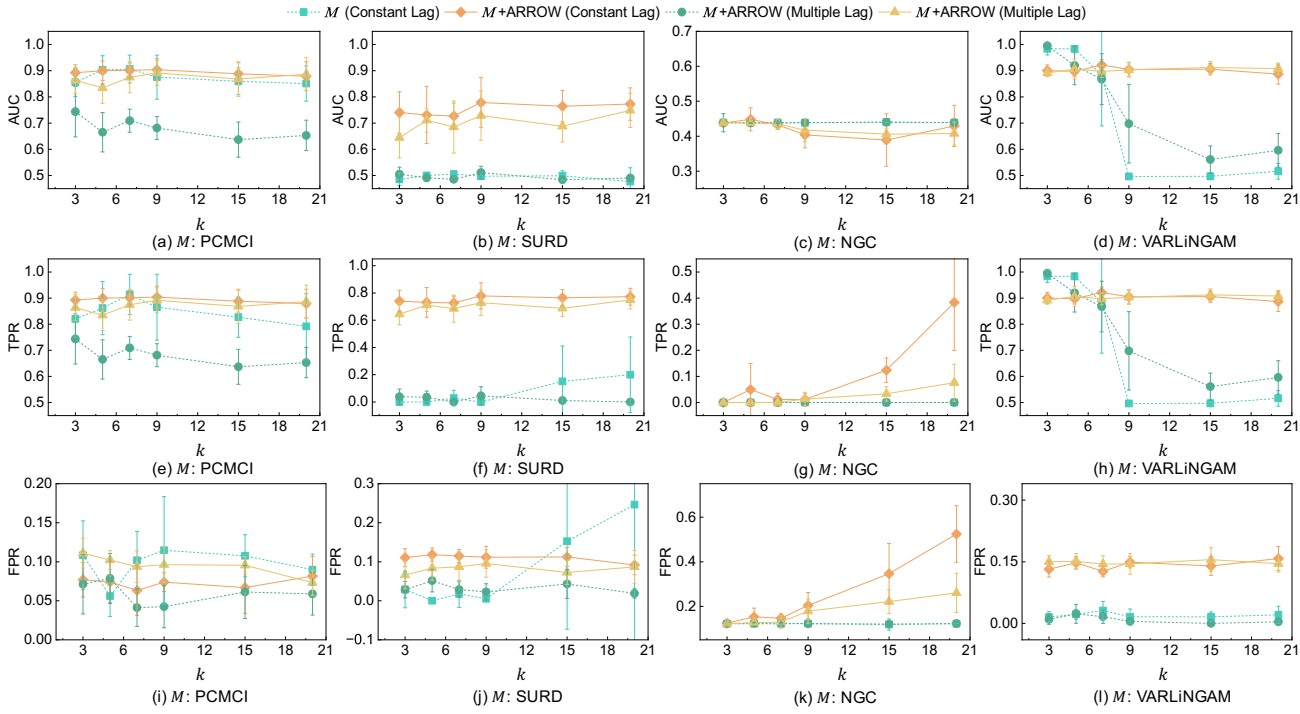

*Figure 9.* Summary graph performance on the dataset with non-linear causal relationships

