# OpenReview forum: "Arrow: Accelerator for Time Series Causal Discovery with Time Weaving"
_ICML.cc/2025/Conference — ICML 2025 poster_

### Official Review · Reviewer_vSZc · 2025-03-10

**Overall Recommendation:** 4

**Summary:**

The authors proposed an accelerator framework named ARROW to address the efficiency bottleneck in multivariate time series causal discovery. By introducing time weaving encoding (capturing contextual trends between time points), an optimal time lag determination theorem based on XOR operations, and an intelligent pruning strategy, ARROW significantly improves causal discovery efficiency without compromising the performance of the original algorithms. Experiments demonstrate that the method is adaptable to various causal discovery algorithms and shows significant improvements in efficiency.

**Claims And Evidence:**

Yes. The claims made in the submission are supported by clear and convincing evidence, including rigorous experimental results, comparative analyses with four different types of methods, and theoretical justifications, all of which collectively validate the effectiveness and efficiency of the proposed ARROW framework.

**Essential References Not Discussed:**

No. The related work section in the paper is comprehensive and thorough.

**Experimental Designs Or Analyses:**

Yes. The experimental designs and analyses in the paper are well-founded and reliable, supported by thorough comparisons with baseline methods in different cases, and consistent results across various datasets, showcasing the robustness and reliability of the proposed approach.

**Methods And Evaluation Criteria:**

Yes. The proposed methods and evaluation criteria, including synthetic datasets and metrics, are well-suited for the problem of causal discovery in multivariate time series. They also demonstrate that ARROW can effectively address computational efficiency and accuracy issues in real-world applications.

**Other Comments Or Suggestions:**

None

**Other Strengths And Weaknesses:**

**Strengths**

S1. The proposed accelerator, ARROW, aims to achieve high efficiency for various existing causal discovery methods.

S2. Experimental results demonstrate that the proposed accelerator, ARROW, successfully addresses the efficiency bottlenecks of existing causal discovery methods, achieving a maximum speedup of 153 times. ARROW is a research with significant practical implications.

S3. The paper is well written.

**Weaknesses**

W1. The paper mentions using XOR operations to determine the optimal time lag, but is the XOR operation applicable to all types of time series data? Are there certain data distributions or causal relationship patterns that could cause the XOR operation to fail?

W2. Compared to existing high-performance causal discovery methods (such as those based on GPU acceleration), what are the advantages of ARROW?

W3. From the experimental results, it appears that ARROW shows more significant acceleration and performance improvement for nonlinear causal relationships. Please provide a detailed explanation.

**Questions For Authors:**

Answer W1, W2, W3.

**Relation To Broader Scientific Literature:**

The paper's key contributions are deeply connected to the broader scientific literature by addressing efficiency bottlenecks in causal discovery, introducing innovative encoding and optimization techniques, and demonstrating practical value through empirical validation, thereby advancing the field and its real-world applications.

**Theoretical Claims:**

Yes. The correctness of the theoretical claims, including the proofs for the XOR-based time lag determination theorem (Appendix A), appears to be well-supported by logical reasoning and empirical validation, ensuring their validity within the proposed framework.

---

> ### Author Rebuttal · Authors · 2025-03-31
>
> We appreciate the insightful comments and our responses are detailed below.
>
> **Response to W1**: Our method is better suited for monotonic causal relationships and less applicable to purely nonlinear ones. Future work should explore trend patterns in nonlinear relationships, such as periodicity. In complex scenarios, some nonlinear relationships don't affect overall monotonicity, and thus, our method remains effective. This is validated by our experiments on synthetic datasets with nonlinear causal relationships, as shown in Table 2 of the paper.
>
> **Response to W2**: GPU-based acceleration provides hardware-level acceleration. In contrast, ARROW is the first data-level acceleration solution, particularly suited for high-dimensional time series data, and can be used alongside GPU acceleration.
>
> **Response to W3**: Nonlinear datasets pose challenges for the downstream causal discovery algorithms. However, by identifying time lags and pruning variables in advance, our approach enhances time lag discovery, causal graph accuracy, and causal mining efficiency.

---

> > ### Comment · Reviewer_vSZc · 2025-04-07
> >
> > I would like to thank the authors for their detailed  rebuttal and confirm to vote for acceptance. Thanks.

---

### Official Review · Reviewer_YSzU · 2025-03-11

**Overall Recommendation:** 5

**Summary:**

This paper presents ARROW, an accelerator for time series causal discovery that overcomes the efficiency bottleneck of existing causal discovery methods. The concept of time weaving is introduced, along with an XOR-based time lag discovery strategy, which leverages theoretical derivations to rapidly determine the optimal time lag, significantly improving computational efficiency. Additionally, a pruning strategy is designed to optimize the search space. The experimental results demonstrate that ARROW significantly accelerates the causal discovery process.

**Claims And Evidence:**

Yes. This paper proposes a general accelerator for time-series causal discovery algorithms. Its effectiveness is validated through theoretical analysis and empirical evaluation across four different types of causal discovery methods.

**Essential References Not Discussed:**

I believe that the paper has sufficiently discussed the works related to its research.

**Experimental Designs Or Analyses:**

Yes. The experimental design and analysis are highly reasonable. To validate the effectiveness of the proposed method, the authors conducted tests on both synthetic and real-world datasets using four different causal discovery algorithms. Additionally, they set up both constant time lags and multiple time lags for the varying relationships between variables in multivariate scenarios. Extensive evaluations demonstrate that ARROW can achieve stable and efficient causal discovery across most causal discovery algorithms.

**Methods And Evaluation Criteria:**

Yes. To improve the efficiency of temporal causal discovery, the paper introduces the concept of time weaving and an XOR-based sequence analysis method, leveraging bit-level operations to accelerate the process. Additionally, the pruning strategy ensures the accuracy of time lag identification. The paper evaluates the performance of both time lag discovery strategy and causal graph construction, demonstrating the method's reliability and effectiveness in causal discovery tasks.

**Other Comments Or Suggestions:**

1. In the sentence "their time lag discovery and causal graph generation performance is assessed using three metrics," "is" should be "are."

**Other Strengths And Weaknesses:**

S1. The proposed accelerator, ARROW, effectively addresses the efficiency bottleneck of existing causal discovery methods. ARROW is with broad applicability and practical value.
S2. The paper introduces a novel concept of "time weaving" and leverages XOR technology for sequence analysis, offering an elegant approach that greatly enhances computational efficiency.
S3. The paper designs an efficient and effective pruning strategy that accurately identifies the most relevant candidate variables, reduces the search space, and quickly determines the optimal time lag. Additionally, the paper provides strong theoretical support.
S4. Experimental results fully validate the effectiveness of the accelerator, demonstrating it superior performance in computational efficiency.

W1. In cases when different time lags exist between multivariate variables, calculating the time lag pairwise may increase the complexity. Would this affect the overall efficiency?
W2. The paper introduces the time weaving concept with a hyperparameter w. It would be better to experimentally verify the impact of this parameter on both efficiency and effectiveness.
W3. The description of handling irregular data is not very clear. If there is a lot of missing data, should the window size be fixed or variable?

**Questions For Authors:**

{W1, W3}

**Relation To Broader Scientific Literature:**

Currently, most causal discovery research focuses on causal discovery strategies in different scenarios, lacking a general accelerator framework to improve the efficiency of causal discovery. This paper primarily addresses the efficiency problem in time series causal discovery and can be applied to most existing causal discovery methods.

**Theoretical Claims:**

Yes. I carefully reviewed the proof section in the appendix, which thoroughly demonstrates Theorem 4.2, and provides an explanation for why the threshold value is set to 0.33 in the pseudocode.

---

> ### Author Rebuttal · Authors · 2025-03-31
>
> We appreciate the positive comments and our responses are detailed below.
>
> **Response to W1**: Calculating time lag is the same for both constant and multiple lags, with no additional complexity for multiple lags. Our pruning strategy reduces the time dimension and variable count, while binary computation improves the efficiency.
>
> **Response to W2**: In the rebuttal, we add a comparison experiment on the hyperparameter w, evaluating the SURD algorithm on a nonlinear dataset with w set to {3, 9, 15, 21}. The results below indicate that larger w values slightly accelerate ARROW without significantly affecting time lag discovery or causal graph accuracy, further highlighting its effectiveness in handling irregular time series data.
>
> | w | Graph AUC | Lag AUC | Time(s) |
> |-----------|-----------|---------|---------|
> | 3         | 0.745     | 0.923   | 6.273   |
> | 9         | 0.686     | 0.886   | 6.253   |
> | 15        | 0.746     | 0.941   | 6.173   |
> | 21        | 0.716     | 0.912   | 6.030   |
>
> **Response to W3**: The window size for handling irregular data is dynamically adjustable, based on the sparsity of the data.

---

### Official Review · Reviewer_yPiC · 2025-03-13

**Overall Recommendation:** 2

**Summary:**

This paper presents ARROW, an acceleration framework for causal discovery in time series data. ARROW aims to improve the efficiency of causal discovery algorithms by reducing computational complexity through three sequential steps: Time Encoding with Time Weaving transforms time series into binary tuple representations to capture local trend dynamics. Time Lag Discovery via XOR Analysis identifies optimal time lags by analyzing trend patterns using XOR operations, ensuring efficient and accurate selection. Candidate Pruning Strategy reduces the search space by filtering out irrelevant variable pairs, improving efficiency without compromising accuracy. ARROW successfully accelerates four time series causal discovery algorithms by up to 153x on 25 synthetic and real-world datasets while improving accuracy in most cases.

################################
Added after rebuttal period:  I checked the authors' response, but I still believe that the assumption in Theorem 4.2 of the paper is too strong. As the authors mention, the method is effective in scenarios with monotonic causal relationships, but in more complex systems, such as financial markets, climate, where there are multi-variable interactions, the relationship between two variables is likely to be influenced by other variables, non-stationarity . This could cause the numerical trend consistency to disappear, rendering the ARROW method ineffective. The authors claim that there are no specific requirements for data construction, but if two variables, A and B, only have a dependency in the first 1/5 of the time and the dependency disappears afterward, I believe it would be difficult for ARROW to detect a significant cooperative positive/negative trend that exceeds the set threshold in this case. In summary, I think the core assumption in Theorem 4.2 has certain limitations, so I keep my original score.

**Claims And Evidence:**

The authors' claims in this paper are theoretically supported, specifically by Theorem 4.2. However, this theorem assumes that if a variable v has a causal effect on v′, then their trends are more likely to be either positively or negatively correlated. I believe this assumption is overly idealized, as real-world systems often involve multiple interacting variables, where the effect of one variable may be offset or influenced by others, making the overall trend less predictable.Under the authors' assumptions, the proposed methods and evaluation criteria are reasonable. However, their applicability to real-world scenarios remains uncertain.

**Essential References Not Discussed:**

I do not see any major issues with the related work. The paper cites relevant prior research in time series causal discovery and acceleration techniques, providing sufficient context for its key contributions.

**Experimental Designs Or Analyses:**

The experiments include both synthetic and real-world experiments. The synthetic experiments consider multiple scenarios and appear to be relatively comprehensive. However, I do not get the information of the number of variables and the edge density in each synthetic dataset from the paper. Additionally, the real-world dataset is not specifically described in detail in the paper. It is also unclear whether the synthetic experiments effectively validate the hypotheses proposed in the paper.

**Methods And Evaluation Criteria:**

Under the authors' assumptions, the proposed methods and evaluation criteria are reasonable. However, their applicability to real-world scenarios remains uncertain.

**Other Comments Or Suggestions:**

1.  Provide more details on the synthetic datasets, including the number of variables and the density of causal graphs, to enhance the transparency of the evaluation.
2.  Clarify the description of the real-world dataset, particularly the specifics of the DREAM3 dataset, to better assess the method’s applicability.
3.  Further discuss the limitations of the approach, especially its potential challenges in unobserved confounders.

**Other Strengths And Weaknesses:**

Strength:
1. The paper is clearly written and presents its methodology in an understandable way.

Weaknesses:
1. I believe the proposed method may not be well-suited for real-world scenarios, as a variable is often influenced by multiple other variables in general. The impact of one variable may be offset by the effects of others, meaning that trend changes do not necessarily exhibit a strictly positive or negative correlation, which contradicts the assumption of Theorem 4.2.

2. The authors do not specify the specific conditions under which their method is applicable, such as whether it assumes stationarity or causal sufficiency. Clarifying these assumptions would help better understand the method’s limitations and applicability.

3. Although the authors claim that their method is not a causal discovery approach but rather a data-level acceleration framework applicable to most causal discovery methods, I believe it fundamentally serves as a causal discovery pre-filtering step. It leverages trend-based analysis to determine whether a time series is a potential causal candidate for another variable. However, I believe this trend-based approach is prone to misclassification. While it may be beneficial in sparse causal graphs, its effectiveness could diminish as the relationships between variables become more complex.

**Questions For Authors:**

1. Theorem 4.2 suggests that if variable v has a causal effect on v′, the probability of the XOR operation results being (0,0) or (1,1) is higher than (0,1) or (1,0), implying that the trends tend to be either consistently similar or consistently opposite. Is there any evidence to support this claim? How do you account for the influence of other variables and noise in this process?

2. Does this method apply to scenarios with hidden variables? Since hidden variables are common in real-world data, I believe they may affect the ARROW process, potentially excluding the correct variables and reducing the effectiveness of the subsequent causal discovery methods.

3. Is there a stationarity assumption in your approach? I believe non-stationary time series could disrupt the ARROW process and affect its performance.

4. How is the pruning threshold in the final step determined? Is there a theoretical basis for it, or is it purely empirical?

**Relation To Broader Scientific Literature:**

This proposed method is based on existing time series causal discovery methods by introducing a candidate variable selection step and a time lag determination strategy. These steps reduce the number of variable pairs that need to be tested, effectively accelerating the causal discovery process.

**Theoretical Claims:**

I checked the proof of Theorem 4.2, and under the assumptions proposed by the authors, the proof seems to be correct. However, I believe that the assumptions may not hold in real-world scenarios. The theorem assumes that if a variable v has a causal effect on v′, their trends should be either positively or negatively correlated. While this might be reasonable in a simplified setting with a single influencing variable, real-world systems are often influenced by multiple interacting factors, making the actual trend relationships more complex.

---

> ### Author Rebuttal · Authors · 2025-03-31
>
> Thanks for your valuable suggestions.
>
> **Response to Q1 and W1**:
>
> 1. **Response to the assumption of Theorem 4.2:** Our method is well-suited for scenarios with monotonic causal relationships, as assumed in Theorem 4.2, where trend changes show consistent positive or negative correlations in most consecutive periods. Many real-world datasets (e.g., NetSim[ICLR23], DREAM3, NetSIM[ICLR20], Turbulence[Nature24]) exhibit monotonic causal relationships, making our method applicable to practical scenarios. Additionally, we incorporate nonlinear relationships in our experiments (Table 2), with results further validating the robustness of our approach.
>
> 3. **Response to a variable is often influenced by multiple other variables.** Our method filters variable pairs without strong correlations to accelerate downstream causal discovery. The downstream casual discovery method can support either the case when a variable is influenced by multiple other variables or the case when a variable is influence by one variable, which is not our focus. In situations where the impact of one variable is offset by others, ARROW will not incorrectly prune the edges in most cases. For example, in the case of 2A(t-1) = C(t) and -2B(t-1) = C(t), with A and B being functions of y=x, the effects of A and B cancel each other out, making C always 0. Here, A’s time weaving encoding is {1, 1, 1}, B’s is {0, 0, 0}, and C’s is {0, 0, 0}, so A XOR C always gives {0, 0, 0, 0}, and A XOR B gives {1, 1, 1, 1}. According to Theorem 4.2, both edges will be retained, with causal relationships determined by downstream algorithms. However, in complex scenarios challenging for downstream algorithms, ARROW may misprune, reflecting a trade-off between accuracy and efficiency. Our method performs well on both synthetic datasets and real-world datasets, which involve multiple variable influences, as shown in Tables 1 and 2 in the paper.
>
> 4. **Response to noise.** Our datasets include both synthetic datasets and real-life datasets used in experiments incorporate the noise. In the synthetic datasets, we set the noise standard deviation to 0.1. In this rebuttal, we further validate the impact of different noise levels on ARROW by setting the noise standard deviations in the nonlinear causal relationship dataset to 0.1, 0.5, and 1. The results below demonstrate that noise has a minimal effect on performance. Additionally, the acceleration efficiency remains stable across these noise levels.
>
> | Parameter | Graph AUC | Lag AUC | Time(s) |
> |-----------|-----------|---------|---------|
> | 1         | 0.842    | 0.895  | 7.166  |
> | 0.5       | 0.870    | 0.911  | 7.047  |
> | 0.1       | 0.846    | 0.894  | 7.055  |
>
>
> **Response to Q2**: The hidden variable problem remains a significant challenge in causal discovery[CSUR23]. Our method focuses on accelerating causal mining algorithms, while handling hidden variables is determined by downstream algorithms (e.g., SURD supports hidden variables, but PCMCI does not). When hidden variables exist, we accelerate the discovery only based on observed variables, thus the acceleration process for observed variables remains unchanged regardless of the number of latent variables. This rebuttal further validates ARROW on a synthetic dataset with 10 observed and 4 hidden variables. The results below confirm its improvements in time lag discovery, causal graph accuracy, and up to 70x speedup.
>
> | Metric        | SURD+ARROW    | SURD          |
> |-------------|---------------|---------------|
> | Graph AUC  | 0.645       | 0.484       |
> | Lag AUC    | 0.869        | 0.502        |
> | Time(s)     | 6.219        | 426.206      |
>
> **Response to Q3 and W2:**
> As an accelerator for causal mining, ARROW makes no specific assumptions about the dataset; these are determined by downstream methods. For instance, LiNGAM assumes linear data generation, non-Gaussian disturbances, and no unobserved confounders, while PCMCI assumes causal sufficiency, faithfulness, and the Markov condition.  For datasets with insufficient causality, ARROW does not prune indirect variables that still show lagged correlations with other observed variables, allowing downstream algorithms to make the final decision.
>
>
> **Response to Q4:**
> The pruning threshold is set based on the sparsity levels in real-world scenarios. Currently, the threshold is set to 0.25, which is set based on experience. A larger threshold retains more edges, reducing efficiency, while a smaller threshold may prune important correlations, impacting accuracy.
>
> **Response to W3:**
> The datasets we generated in our experiments have sparsity levels of 0.2 and 0.4, while the real-world datasets, including the Dream3 dataset, have sparsity levels ranging from 0.1 to 0.25, and the NetSim dataset has a sparsity level of 0.14 (see the table below).
>
> | Dataset        | Sparsity |
> |--------------------|----------|
> | Dream3-Ecoli1      | 0.11     |
> | Dream3-Yeast2      | 0.25     |
> | Netsim             | 0.14     |

---

### Official Review · Reviewer_t36X · 2025-03-13

**Overall Recommendation:** 4

**Summary:**

The paper investigates the computational efficiency of causal discovery in multivariate time series. Existing methods face high computational costs when applied to large-scale data, primarily due to issues such as data binning, time lag selection, and candidate set explosion. To address these challenges, the authors propose ARROW, a method designed to accelerate time series causal discovery. ARROW optimizes time lag selection using time weaving encoding and XOR operations while reducing computational overhead through a pruning strategy. Experimental results demonstrate that ARROW significantly improves efficiency while maintaining the accuracy of causal discovery.

## Update after rebuttal: I have read the rebuttal and I think my concerns are well addressed, I will keep my score this time (for acceptance).

**Claims And Evidence:**

Yes. The claims made in the submission are supported by clear and convincing evidence, particularly through experiments on synthetic and real-world datasets across various time lag scenarios.

**Essential References Not Discussed:**

The author provides a comprehensive and thorough review of related work.

**Ethical Review Concerns:**

NA.

**Ethics Expertise Needed:**

["Other expertise"]

**Experimental Designs Or Analyses:**

Yes. The experiment validates ARROW using synthetic and real-world datasets across different time lag scenarios, which is quite convincing.

**Methods And Evaluation Criteria:**

Yes. The proposed methods and evaluation criteria are suitable for addressing the problem and validating the approach.

**Other Comments Or Suggestions:**

NA

**Other Strengths And Weaknesses:**

S1. The proposed acceleration framework ARROW has significant value and partially addresses the efficiency bottleneck in causal discovery.

S2. The paper introduces time weaving representation, which represents trends between three consecutive points using a compact binary format, reducing discretization costs and capturing dynamic time series features.

S3. To optimize time lag selection, it leverages XOR operations to analyze trend patterns, eliminating brute-force search and improving efficiency and reliability.

S4. A pruning strategy for candidate sets is proposed to select only the most causally relevant variables, mitigating candidate set explosion and enhancing scalability.

S5. The experiment is robust, thoroughly examining various time lag scenarios and validating the approach on both synthetic and real-world datasets.

W1. Since time weaving encoding retains only trend information while discarding exact numerical changes, it requires further explanation for the observation that our causal discovery performance in the experiment surpasses the original algorithm.

W2. The authors need further clarification whether ARROW's time weaving representation might perform poorly on stationary sequences or high-noise data.

W3. The experimental setting does not explicitly mention the data scale. It would be better to see the impact of different dataset sizes on the performance of the ARROW method.

**Questions For Authors:**

Please See W1, W2, W3.

**Relation To Broader Scientific Literature:**

Building on prior constraint-based, score-based, granger-based, and information-theoretic methods, ARROW enhances time series causal discovery by introducing time weaving encoding and XOR-based analysis to optimize time lag selection and reduce candidate set complexity, significantly improving efficiency without compromising accuracy.

**Theoretical Claims:**

Yes. The theoretical proof regarding the time lag selection strategy has been checked, with a rigorous logic that effectively validates Theorem 4.2.

---

> ### Author Rebuttal · Authors · 2025-03-31
>
> We appreciate the positive comments and our responses are detailed below.
>
> **Response to W1**: Causal relationships rely more on trend synchronization than numerical transmission. Performance improvements come from three aspects:
> * Noise Robustness – Binary encoding filters noise and highlights structural causal patterns.
> * Lag Discovery – XOR operations accurately identify lagged causal relationships and optimize time lag selection.
> * Pruning Strategy – Pruning reduces the search space.
>
> **Response to W2**: ARROW performs well in stationary sequences and noise scenarios.
> * In stationary sequences, binary encoding of temporal weaving stabilizes, and XOR results remain similar, preventing the pruning. Note that, causal discovery relies on the downstream inference, and ARROW does not impact the accuracy of downstream algorithm.
> * Temporal weaving filters high-frequency noise, but in high-noise environments (e.g., outliers), pre-processing is needed to reduce interference and improve causal identification accuracy.
>
> **Response to W3**: We have included different data sizes, including synthesized datasets with 10 variables and 1000 time points, Dream3 dataset with 10 genes and 84 time points.

---

> > ### Comment · Reviewer_t36X · 2025-04-03
> >
> > The reviewer thanks authors for their rebuttal and confirms the vote for acceptance. Thanks.

---

### Decision · Program_Chairs · 2025-05-01

**Decision:**

Accept (poster)

**Comment:**

This paper received four reviews, of which three were highly positive and one leaned negative. The paper explores the computational efficiency of causal discovery in multivariate time series. Reviewers generally found the proposed acceleration framework, ARROW, to be valuable and well-supported by robust experiments. However, one reviewer raised concerns about the applicability of the method to real-world scenarios. The authors have addressed these concerns in their rebuttal.

As the AC, I agree with the majority of the reviewers. The paper tackles an interesting and important problem, and proposes a convincing and well-executed solution. I therefore recommend acceptance.